# A Comprehensive Survey of Multimodal LLMs for Scientific Discovery

## Abstract

Recent advances in artificial intelligence (AI), especially large language models, have accelerated the integration of multimodal data in scientific research. Given that scientific fields involve diverse data types, ranging from text and images to complex biological sequences and structures, multimodal large language models (MLLMs) have emerged as powerful tools to bridge these modalities, enabling more comprehensive data analysis and intelligent decision-making. This work, $S^3$-Bench, provides a comprehensive overview of recent advances in MLLMs, focusing on their diverse applications across science. We systematically review the progress of MLLMs in key scientific domains, including drug discovery, molecular & protein design, materials science, and genomics. The work highlights model architectures, domain-specific adaptations, benchmark datasets, and promising future directions. More importantly, we also conducted benchmarking evaluations of open-source models on several highly significant tasks, such as molecular property prediction and protein function prediction. Our work aims to serve as a valuable resource for both researchers and practitioners interested in the rapidly evolving landscape of multimodal AI for science.

## 1 Introduction

Recent breakthroughs in artificial intelligence (AI) have been driven by foundation models—large-scale neural networks trained on broad data that can be adapted to diverse tasks (OpenAI, 2023; Grattafiori et al., 2024). In particular, large language models (LLMs) based on the Transformer architecture (Vaswani et al., 2017) have achieved remarkable proficiency in natural language processing, exhibiting emergent abilities such as few-shot learning (Alayrac et al., 2022; Brown et al., 2020; Wei et al., 2021; Kojima et al., 2022; Wei et al., 2022) and human-aligned dialogue generation (Ouyang et al., 2022; Ziegler et al., 2019; Glaese et al., 2022). However, these advances remain confined to text-based inputs and outputs, whereas scientific problems are inherently multimodal—spanning modalities such as clinical text, biomedical images, molecular structures, and genomic sequences, among others (Li et al., 2023b; Luo et al., 2023a; Liu et al., 2025d; Dhanasekar et al., 2025). This has catalyzed a new generation of multimodal large language models (MLLMs) designed to bridge diverse data modalities and enable more comprehensive reasoning.

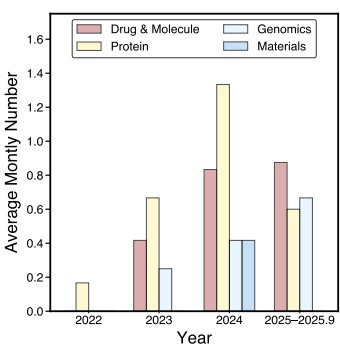

Figure 1: Average monthly number of publications on MLLMs in science (2022–present), collected from arXiv, Nature, and bioRxiv, showing the increasing attention to MLLM applications in science.

MLLMs extend language modeling beyond text, enabling AI systems to ingest and generate diverse data types such as images, audio, and structured scientific representations (Yin et al., 2024; Wu et al., 2023; Liang et al., 2024b). Early examples like Flamingo (Alayrac et al., 2022) and Kosmos-1 (Huang et al., 2023) showed that LLMs can be adapted or trained to jointly reason over visual and textual inputs, while open-source efforts such as MiniGPT-4 (Zhu et al., 2023) and LLaVA (Li et al., 2023a) align vision encoders with LLMs, marking a shift from text-only AI towards generalist multimodal agents. This multimodal trend is especially impactful in science, where tasks often integrate multiple modalities. Biomedical models such as BioMedGPT (Luo et al., 2023a) unify

protein sequences, molecular structures, and textual knowledge for drug discovery. In genomics, systems like Geneverse (Liu et al., 2024e) and GeneChat (Dhanasekar et al., 2025) connect DNA sequences with biomedical knowledge. In materials science, multimodal AI can parse literature and microstructure images jointly to propose new materials or predict properties (Boyar et al., 2025; Buehler, 2024; Alampara et al., 2024; Pyzer-Knapp et al., 2025). Across these domains, MLLMs act as engines that fuse language with domain-specific modalities, enabling holistic analysis and accelerating discovery (Figure 1).

Given this rapid progress, there is a pressing need to systematically survey MLLMs in science. Existing surveys mainly focus on general-purpose LLMs (e.g., (Zhao et al., 2023)) or on narrower multimodal techniques (e.g., (Yin et al., 2024)). Domain-specific reviews exist for biology or biomedicine (Zhang et al., 2024c; 2025b; Thirunavukarasu et al., 2023; Zhou et al., 2023a; He et al., 2025; Xiao et al., 2024a; Zheng et al., 2025; Liu et al., 2024b; Wang & Zhang, 2024; Wang et al., 2025d), but no prior work offers a unified overview across natural language, biomedical imaging, molecular data, genomics, and material science (Table 1).

To fill this gap, we present $S^3$-Bench, a comprehensive study with benchmarking evaluation of MLLMs for scientific discovery. Our contributions are threefold: (1) We present the first comprehensive survey work of MLLMs across major scientific domains—including drug discovery, protein engineering, genomics, materials science, and biomedicine—highlighting representative model architectures, domain-specific adaptations, and benchmark datasets. (2) we synthesize emerging directions, including diffusion-based LLMs and multimodal diffusion-based LLMs, and outline open challenges for future research (Appendix C); and (3) we conduct benchmarking experiments on selected open-source MLLMs, evaluating their performance

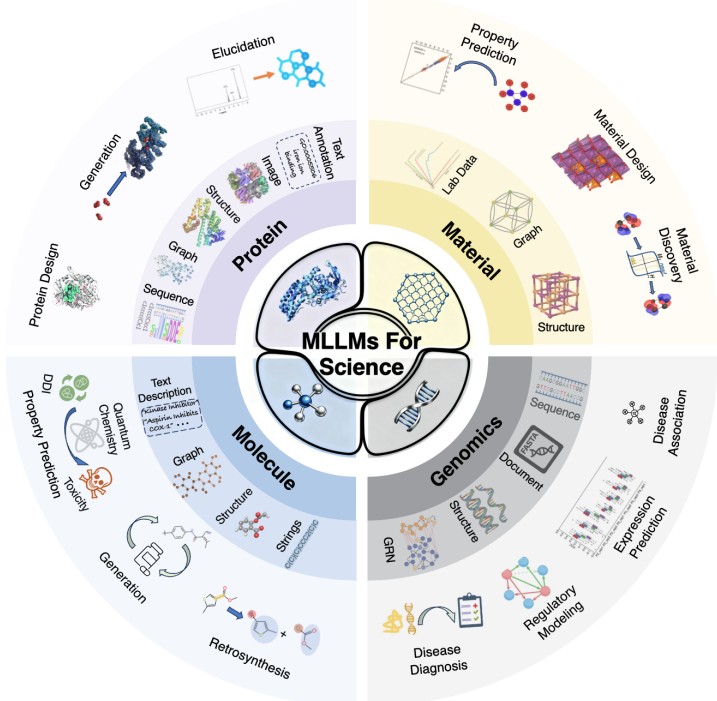

Figure 2: Overview of our $S^3$-Bench, highlighting four major components discussed in the paper and presenting the key modalities and their corresponding applications in this field.

on highly significant tasks such as molecular property prediction and protein function prediction (Appendix D). In summary, MLLMs are rapidly evolving and hold immense promise for advancing scientific discovery, by consolidating progress across diverse modalities and domains and by providing empirical benchmark results, this survey aims to serve as both a reference and a foundation for future work. The paper is organized as follows: Sections 3–6 review domain-specific developments. We also discuss emerging topics and future directions in Section 7.

## 2  DATA REPRESENTATIONS FOR MOLECULES, PROTEINS, AND GENOMES

Across recent work, several families of data representations are used, each with clear trade-offs. For **2D small molecules**, linear strings such as SMILES are compact, human-readable and directly compatible with NLP architectures, but they are non-unique, syntactically fragile (most random strings are invalid) and do not explicitly encode 3D geometry or phenomena like resonance and tautomerism. SELFIES retains the convenience of string representations while enforcing 100% chemical validity by construction, which greatly stabilizes generative models, but yields longer,

less interpretable sequences and still does not solve all issues around stereochemistry or chemically uninteresting-yet-valid structures. IUPAC names provide standardized, human-interpretable nomenclature that aligns well with chemists' intuition, yet they are verbose, morphologically irregular and not trivially invertible to unique structures, making them relatively cumbersome for machine learning despite some recent IUPAC-based models. For **3D molecular structures**, Cartesian atom coordinates give an exact, general description of conformations that integrates naturally with physics-based methods, but they are high-dimensional, sensitive to rigid-body transformations and do not by themselves encode bonding patterns; in contrast, internal/torsional-angle representations (bond lengths, angles and dihedrals) are more compact, chemically intuitive and rotation/translation invariant, yet depend on a predefined molecular graph, struggle with rings and topological changes, and must be converted back to Cartesian space for many downstream computations. For **proteins**, sequence-based encodings (one-hot vectors or embeddings from protein language models) are simple, scalable and benefit from massive sequence databases, but they only capture structure indirectly, whereas structural encodings—such as distance/contact maps, backbone torsion angles, residue-level graphs or discrete structural alphabets—make 3D relationships explicit and often improve structure-aware tasks, at the cost of requiring experimental or predicted structures and increasing representation complexity. Finally, **genomic data** are tokenized either at the character level (A/C/G/T, which is lossless but yields very long sequences), via $k$-mers (which capture local motifs and shorten sequences but blow up the vocabulary and can blur reading frames), or using subword/BPE or hybrid $k$-mer+BPE schemes, which compress frequent patterns and improve language-model efficiency but may segment biologically meaningful motifs in model-dependent ways and can introduce distribution-specific biases.

## 3  MLLMs for Molecule Science and Drug Design

Multimodal large language models (MLLMs) are transforming molecular science and drug discovery by combining different chemical representations such as SMILES (1D) (Weininger, 1988), SELFIES (1D) (Krenn et al., 2020), molecular graphs (2D) (Duvenaud et al., 2015) and geometric structure (3D) (Golkov et al., 2020). They improve key tasks including property prediction, molecular generation, reaction planning, and synthesis optimization, thus accelerating the discovery of novel compounds. In this section, we review recent progress along four directions: (1) LLMs for molecular representation and design, focusing on SMILES- and graph-based embeddings as well as generative models; (2) MLLMs for 1D and 2D tasks, where string and graph/image representations are fused; (3) MLLMs with 3D integration, which enhance structural understanding and retrosynthesis; and (4) chemistry-focused agents and specific applications, covering tool-augmented systems, puzzle-style reasoning, and reaction optimization. Table E1, Table F1, Table F2 and Figure 3 summarize models, datasets, and the research landscape. We also present the benchmarking results of molecular property prediction in Appendix D.

### 3.1  LLMs for Molecule Representation and Design

While our work centers on multimodal LLMs, we also include an overview of LLMs for molecular science to give readers a comprehensive understanding of progress in this field. LLMs are advancing molecular science by learning from diverse chemical representations (Wigh et al., 2022), including the aforementioned 1D, 2D, and 3D data. Transformer models such as ChemBERTa (Chithrananda et al., 2020) and MolBERT (Fabian et al., 2020)

Table 1: Comparison of coverage of recent survey papers on LLMs/MLLMs across different domains.

| Survey | Protein | Drug & Samll Molecule | Gene | Material | Biomedicine | Target Multimodal | Benchmarking |
|---|---|---|---|---|---|---|---|
| **Our Survey** | ✓ | ✓ | ✓ | ✓ | ✓ | ✓ | ✓ |
| *LLMs/MLLMs for Science* | | | | | | | |
| Zhang et al. (2024c) | ✓ | ✓ | ✓ | ✓ | | | |
| Zhang et al. (2024b) | ✓ | ✓ | ✓ | | | | |
| Hu et al. (2025b) | ✓ | ✓ | ✓ | ✓ | ✓ | | |
| Chakraborty et al. (2025) | | ✓ | | | ✓ | | |
| *LLMs/MLLMs for Biomedicine* | | | | | | | |
| Xiao et al. (2025a) | | | | | ✓ | | |
| Ye & Tang (2025) | | | | | ✓ | ✓ | |
| Wang et al. (2024a) | | ✓ | | | ✓ | | |
| Zhou et al. (2023a) | | | | | ✓ | | |
| Buess et al. (2025) | ✓ | | ✓ | | ✓ | | |
| Zheng et al. (2025) | | | | | ✓ | | |
| Liu et al. (2024b) | | | | | ✓ | | |
| He et al. (2025) | | | | | ✓ | | |
| Xiao et al. (2024a) | | | | | ✓ | | |
| Wang et al. (2025d) | | | | | ✓ | | |
| Wang & Zhang (2024) | | | | | ✓ | | |
| Thirunavukarasu et al. (2023) | | | | | ✓ | | |

yield rich embeddings that improve property, drug-target, and drug-drug interaction prediction (Honda et al., 2019; Jin et al., 2025). For de novo design, models like MolGPT (Bagal et al., 2021), ChatMol (Zeng et al., 2024), and ChatDrug (Liu et al., 2024f) generate valid and novel compounds via conditional generation, reinforcement learning, or molecular editing (Chenthamarakshan et al., 2020). LLMs further support multi-objective optimization and iterative refinement with expert or oracle feedback (Wu et al., 2024b). In reaction prediction and synthesis, the *Molecular Transformer* excels in forward and retrosynthetic tasks (Liu et al., 2017), while multimodal and

instruction-following models bridge chemical language with experimental reasoning (Tetko et al., 2020). Overall, LLMs are emerging as powerful engines for molecular discovery, optimization, and synthesis.

## 3.2 MLLMs for 1D and 2D Molecular Tasks

Recent advances in molecular AI highlight a fundamental paradigm shift from single-modality models toward deeply integrated MLLMs, particularly focusing on the fusion of 1D (e.g., SMILES, SELFIES) and 2D (e.g., molecular graphs, structure images) representations (Bhattacharya et al., 2024; Rollins et al., 2024; Jin et al., 2025; Lee et al., 2025; Hu et al., 2024; Le et al., 2024; Deng et al., 2024; Zhang et al., 2024a; Li et al., 2025b; Liu et al., 2024c; Tran et al., 2024; Chen et al., 2025b; Livne et al., 2024; Cao et al., 2023; Luo et al., 2023b; 2022). This shift is motivated by the realization that 1D string representations provide scalability and access to abundant chemical databases, but alone cannot capture the rich spatial, topological, and functional information encoded in 2D modalities. Early progress in the field centered around models leveraging 1D molecular strings, but these were soon recognized as insufficient for tasks demanding a nuanced understanding of molecular connectivity and spatial arrangement. Addressing this, recent works such as Mol-PROP (Rollins et al., 2024) pioneered the fusion of pretrained language models with GNN-based graph encoders, achieving significant gains in property prediction. This line of research has since been extended by LLM-MPP (Jin et al., 2025), Mol-LLM (Lee et al., 2025), and related models such as M³LLM (Hu et al., 2024), which employ advanced architectural innovations such as cross-attention between SMILES, molecular graphs, and textual descriptions, large-scale instruction tuning, and multi-level graph feature integration, resulting in strong and generalizable performance across property prediction, reaction, and generation tasks. Modular and adapter-based approaches, including MolX (Le et al., 2024) and ChemLML (Deng et al., 2024), make it possible to flexibly combine graph encoders with LLMs and rapidly adapt to new tasks with minimal parameter overhead. Meanwhile, tokenizer-based solutions like UniMoT (Zhang et al., 2024a) unify 1D and 2D information at the token level, enabling seamless molecule-to-text and text-to-molecule generation. Beyond graph representations, vision-enhanced models such as ChemVLM (Li et al., 2025b), GIT-Mol (Liu et al., 2024c), and Mol2Lang-VLM (Tran et al., 2024) incorporate 2D structure images alongside textual and graph modalities, further boosting captioning and molecular understanding. On the system level, frameworks like ModuLM (Chen et al., 2025b) and nach0 (Livne et al., 2024) generalize the multimodal paradigm by supporting arbitrary combinations of 1D, 2D, and even 3D encoders, while InstructMol (Cao et al., 2023) and BioMedGPT (Luo et al., 2023b) demonstrate the value of multi-stage instruction tuning and domain-specific integration for high-stakes biomedical applications. Importantly, domain-specialized models such as BioGPT (Luo et al., 2022) represent a milestone in biomedical molecular research. Pre-trained on large-scale PubMed literature, BioGPT achieves state-of-the-art results in biomedical text generation and knowledge extraction, accelerating automated molecular discovery from unstructured data. Collectively, these studies demonstrate that fusing 1D and 2D modalities not only consistently improves accuracy and generalizability for property prediction, generation, and retrosynthesis tasks, but also lowers the barrier for extending models to new modalities and domains. As such, the evolution from 1D-only to 1D&2D-fused MLLMs marks a major leap for molecular AI, setting a new foundation for interpretable, robust, and transferable molecular representation learning in chemistry, biology, and drug discovery.

## 3.3 MLLMs with 3D Geometry Integration for Molecular Tasks

Recent advances in MLLMs with 3D geometry integration can be broadly categorized by their target molecular tasks. For *representation learning and property prediction*, MolBind (Xiao et al., 2024c) aligns scientific language, 2D molecular graphs, 3D conformations, and protein pockets into a unified representation space via contrastive learning, enabling cross-modal retrieval and zero-shot molecular property prediction. Similarly, ModuLM (Chen et al., 2025b) provides a modular framework that flexibly combines 1D, 2D, and 3D encoders with diverse LLM backbones, facilitating benchmarking and adaptation across a wide range of molecular tasks. BindGPT (Zholus et al.,

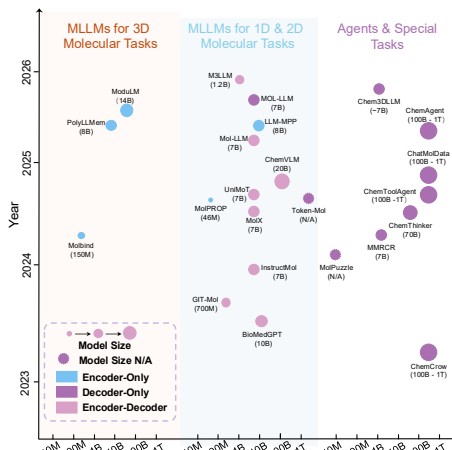

Figure 3: Distribution of MLLMs for drug and molecule tasks, presenting each model's release date, scale, architecture and applica-

2025) proposes a large-scale pre-trained language-model–based framework, further optimized with reinforcement learning, that can efficiently generate 3D molecular structures—including pocket-conditioned ligands, conformers, and unconditional molecules—for structure-based drug design. 3D-MoLM (Li et al., 2024b) equips a language model with a 3D molecular encoder and a 3D molecule-text projector plus instruction tuning so that it can interpret 3D molecular structures and achieve strong performance on molecule-text retrieval, captioning, and open-text molecular QA, especially for 3D-dependent properties. 3DSMILES-GPT (Golkov et al., 2020) is a fully token-only large language model framework that treats both 2D and 3D molecular information as sequences, then pretrains, pocket-conditions, and RL-fine-tunes this model to rapidly generate 3D drug-like molecules in protein pockets with improved binding affinity, drug-likeness, and synthesizability over prior methods. For *reaction modeling*, RetroInText (Kang et al., 2025) integrates 3D geometry, 2D molecular graphs, and in-context reaction text to enhance multi-step retrosynthesis, particularly for long and complex synthetic routes. For *materials and polymer science*, PolyLLMem (Zhang & Yang, 2025) couples Llama3-based SMILES embeddings with Uni-Mol 3D embeddings through a gated fusion mechanism, demonstrating strong performance in polymer property prediction under limited-data scenarios. Overall, these approaches reflect a growing trend toward fully multimodal MLLMs that combine complementary molecular representations (1D, 2D, and 3D) to achieve improved accuracy, interpretability, and generalizability across chemical and biological domains

### 3.4 MLLMs for Chemistry-Focused Agents and Special Applications

(1) *Chemistry-Focused Agents.* Recent work has introduced chemistry-focused agents that couple MLLMs with domain-specific tools to automate molecular data processing and reasoning (Bran et al., 2023; Yu et al., 2024b;a; Tang et al., 2025; Ju et al., 2024). Examples include ChatMol-Data (Yu et al., 2024b), which integrates modules for literature mining, structure handling, and database operations; ChemCrow (Bran et al., 2023) and ChemToolAgent (Yu et al., 2024a), which enhance LLMs for synthesis planning and property prediction; and ChemAgent (Tang et al., 2025) and ChemThinker (Ju et al., 2024), which introduce memory or multi-agent designs for more accurate and interpretable reasoning. (2) *Puzzle and Reaction Condition Recommendation.* Beyond standard benchmarks, chemistry also involves expert-level reasoning tasks that require integrating diverse data sources. Puzzle-style problems (Navaza & Saludjian, 1997; Adams et al., 2011; Zwart et al., 2008; Fricker et al., 2004; Bunkóczi et al., 2013), such as structure elucidation from spectroscopic clues, test the limits of MLLMs; MolPuzzle (Guo et al., 2024) shows that while models like GPT-4o handle simple cases, they still lag behind human experts. Similarly, tasks such as reaction condition recommendation and synthesis optimization demand advanced reasoning. MM-RCR (Zhang et al., 2024d) exemplifies progress here by unifying textual, graph, and SMILES data, achieving state-of-the-art results and strong generalization. Overall, MLLMs are moving from uni-modal to fused 1D/2D/3D, agent-augmented systems that boost property prediction, generation, retrosynthesis, and condition recommendation. We believe key hurdles remain in rigorous reasoning, interpretability/reproducibility, and closed-loop experimental and safety integration.

## 4 MLLMs for Protein Science

As protein-related tasks increasingly involve diverse data modalities, including natural language descriptions (1D) , amino acid sequences (1D) , protein graph (2D) , and protein geometric structures (3D) , MLLMs have emerged as a powerful framework for integrating these heterogeneous sources of information (Liu et al., 2025d; Hayes et al., 2025; Zhou et al., 2025c). Unlike unimodal models, MLLMs can jointly reason across multiple biological representations, enabling more expressive learning and flexible interaction with biological data. In this section, we review recent advances in MLLMs across three major categories: (1) we examine models that integrate protein sequences with textual information, supporting tasks such as protein captioning, design, and function prediction. (2) we discuss models that incorporate geometric representations alongside sequence and text, enabling structure-aware learning for enhanced prediction and generation. (3) we highlight MLLMs developed for specialized tasks, including protein–protein and free-text-based biological translation. Table E2, Table F3, Table F4 and Figure 4 summarize models, datasets, and the research landscape. We also present the benchmarking results of protein function prediction in Appendix D.

## 4.1 LLMs FOR PROTEIN SCIENCE

We likewise begin by providing an overview of LLMs in protein science for readers to contextualize the broader advances in this domain. Large language models have revolutionized protein science, enabling efficient and scalable solutions for major challenges in protein property prediction, function annotation, structure prediction, and protein engineering (Alley et al., 2019; Elnaggar et al., 2021; Rives et al., 2021; Jumper et al., 2021; Madani et al., 2023a). In property prediction, models such as UniRep (Alley et al., 2019) and ProtTrans (Elnaggar et al., 2021) leverage large-scale pretraining to achieve state-of-the-art accuracy on tasks including stability, solubility, and fluorescence. For function annotation, transformer-based models like ESM-1b (Rives et al., 2021), MSA Transformer (Rao et al., 2021), TCR-BERT (Wu et al., 2024a), and ProteinBERT (Brandes et al., 2022) have significantly improved label prediction, enzyme classification, and TCR-antigen binding. In structure prediction, advances such as AlphaFold2 (Jumper et al., 2021), ESMFold (Lin et al., 2023b), and ESM-IF (Hsu et al., 2022) have enabled end-to-end and inverse folding, approaching experimental-level 3D accuracy. Models like GearNet (Zhang et al., 2023), SaProt (Su et al., 2023b), and OntoProtein (Zhang et al., 2022c) integrate structural knowledge and ontologies, further enhancing performance on structure-aware tasks. For protein engineering and generation, ProGen (Madani et al., 2023a), ProtGPT2 (Ferruz et al., 2022), and ProGen2 (Nijkamp et al., 2023) apply autoregressive and conditional generation to produce novel, functional, and diverse proteins. Specialized models such as IgLM (Shuai et al., 2023) and PALM-H3 (He et al., 2024) address antibody and virus-specific design. Collectively, these advances establish Protein LLMs as powerful engines for biological discovery and rational protein design, expanding the reach of AI-driven protein science (Rives et al., 2021; Jumper et al., 2021; Madani et al., 2023a; Brandes et al., 2022; Lin et al., 2023b).

## 4.2 MLLMs FOR PROTEIN SEQUENCE–LANGUAGE INTEGRATION

Recent advancements in MLLMs that integrate protein sequences with textual descriptions have led to significant progress in protein-related tasks (Liu et al., 2025d; 2024g; Zhou et al., 2025a; Dharuman et al., 2024; Zhang et al., 2022a; Luo et al., 2023a; Zhuo et al., 2024; Lv et al., 2025; Wang et al., 2023a; Li et al., 2023d; Zheng et al., 2024; Pei et al., 2023; 2024; Taylor et al., 2022; Wang et al., 2023b; Huo et al., 2024; Zhou et al., 2025c; Chen et al., 2024a). ProteinDT (Liu et al., 2025d) combines protein sequences with textual prompts for protein design, achieving high accuracy in generating novel proteins. ProtT3 (Liu et al., 2024g) excels in generating text descriptions from protein sequences using a Q-Former encoder, specifically targeting protein captioning and QA tasks. ProtCLIP (Zhou et al., 2025a) enhances protein function prediction by integrating protein sequences with textual knowledge graphs, further improving prediction accuracy. BioMedGPT (Luo et al., 2023a) expands this by incorporating both protein sequences and textual knowledge for biomedical question answering, enabling improved understanding and reasoning in the biomedical domain. PROTLLM (Zhuo et al., 2024) and ProLLaMA (Lv et al., 2025) bridge protein sequence understanding and generation tasks, with ProLLaMA excelling in multi-task learning, particularly in protein structure and function prediction. InstructProtein (Wang et al., 2023a) aligns protein sequences with natural language through knowledge-guided instructions, improving task handling.

Other models such as DrugGPT (Li et al., 2023d) and ESM-AA (Zheng et al., 2024) target drug design and molecular modeling, tackling ligand generation and protein interaction analysis. BioT5 (Pei et al., 2023) and BioT5+ (Pei et al., 2024) integrate molecular properties with text for multi-task protein understanding. OntoProtein (Zhang et al., 2022a) fuses Gene Ontology with sequences to improve function prediction (e.g., GO-CC/GO-BP). Galactica (Taylor et al., 2022) trains on a curated scientific corpus for multimodal reasoning, outperforming GPT-3 on LaTeX and PubMedQA. For multimodal protein tasks, BioBRIDGE (Wang et al., 2023b) links unimodal biomedical models via knowledge graphs to predict drug–target and protein–protein interactions. xTrimoPGLM (Chen et al., 2024a) unifies protein understanding and generation, achieving state-of-the-art results. ProteinChat (Huo et al., 2024) conditions on sequences and text prompts to describe protein functions in free-form and classification settings. LLaPA (Zhou et al., 2025c) combines sequences, PPI networks, and instructions for multi-label PPI and multi-protein affinity prediction. Lastly, MProt-DPO (Dharuman et al., 2024) employs Direct Preference Optimization to surpass the ExaFLOPS barrier in protein design, improving efficiency. Collectively, these models showcase the power of MLLMs that couple sequences with text for protein design, function prediction, and interaction analysis.

### 4.3 MLLMs for Protein Structure–Sequence–Language Integration

Given the critical role of geometric information in understanding protein behavior, recent research has increasingly focused on integrating structural modalities into MLLMs (Hayes et al., 2025; Wang et al., 2024b; Gao et al., 2025; Li et al., 2024a; Lin et al., 2023a; Su et al., 2023a; Wang et al., 2025a; Xiao et al., 2024d; 2025b; Zhuang et al., 2024; Zhou et al., 2025d; Ruffolo et al., 2024). Several representative models—including ESM3 (Hayes et al., 2025), DPLM2 (Wang et al., 2024b), FoldToken (Gao et al., 2025), ProTokens (Lin et al., 2023a), Saprot (Su et al., 2023a), and ProSST (Li et al., 2024a)—incorporate protein structural information using various tokenization strategies. Compared to other models, ESM3 (Hayes et al., 2025) incorporates additional functional tokens designed to support specific protein function design tasks. DPLM2 (Wang et al., 2024b) leverages a GVP-based encoder and an IPA-based decoder to learn structural tokens, fine-tuned from DPLM (Wang et al., 2024c), and achieves strong performance in generative tasks.

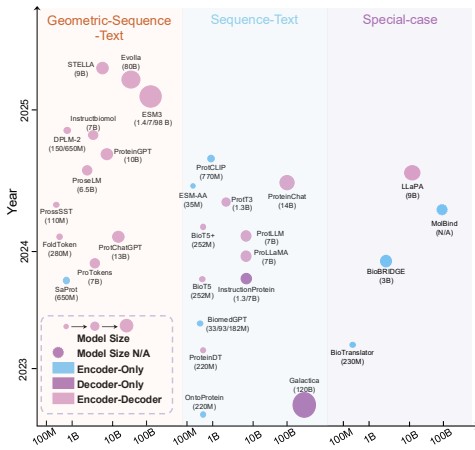

Figure 4: Distribution of MLLMs for protein tasks, presenting each model's release date, scale, architecure and application.

ProTokens (Lin et al., 2023a) employs an SE(3)-invariant transformer to obtain latent structural representations, which are then quantized into discrete tokens that capture structural features. FoldToken (Gao et al., 2025), identifies the limitations of classical quantization approaches and proposes three custom-designed quantizers, whose effectiveness is validated through experimental evaluation. Saprot (Su et al., 2023a) constructs structure-aware tokens with the aid of Foldseek (van Kempen et al., 2022) and performs well across various downstream tasks. ProSST (Li et al., 2024a) differs from previous models by constructing a local structure codebook that captures contextual information beyond individual residues and introducing a sequence–structure disentangled attention mechanism, which is validated through ablation studies.

Beyond tokenization-based approaches, other MLLMs integrate structural information primarily through encoders and align the resulting representations with corresponding sequences or textual data. Models such as ProtChatGPT (Wang et al., 2025a), ProteinGPT (Xiao et al., 2024d), STELLA (Xiao et al., 2025b), InstructBioMol (Zhuang et al., 2024), Evolla (Zhou et al., 2025d), and ProseLM (Ruffolo et al., 2024) exemplify this strategy. The overall architectures of ProtChatGPT (Wang et al., 2025a), STELLA (Xiao et al., 2025b), InstructBioMol (Zhuang et al., 2024), and ProteinGPT (Xiao et al., 2024d) are similar, as they all utilize protein structure encoders. However, ProtChatGPT uniquely incorporates a second protein structure encoder to enhance structural feature extraction, while InstructBioMol adds an additional molecular encoder to integrate molecular information. ProseLM (Ruffolo et al., 2024) employs a causal encoder that integrates structural and functional contexts, successfully designing a PD-1 binder with a binding affinity of 2.2 nM. Evolla (Zhou et al., 2025d) also integrates structural information through protein encoders; however, its distinguishing feature is the use of Direct Preference Optimization (DPO) (Rafailov et al., 2023) as a post-pretraining method. The model is primarily designed for protein-related question answering tasks. SLMs (Lu et al., 2024) encode protein structures into discrete latent tokens with a VQ-VAE and then apply conditional language modeling (including the ESMDiff masked-diffusion model) to efficiently generate diverse, sequence-conditioned protein conformations that are 20–100× faster to sample than traditional 3D diffusion-based methods.

### 4.4 MLLMs for Protein Interactions and Specialized Applications

Understanding protein–protein interactions (PPIs) (Nooren & Thornton, 2003) is critical for elucidating protein function, and several MLLMs have been developed for this task. LLaPA (Zhou et al., 2025c) integrates protein and graph encoders with a language model in a multimodal fusion framework, while BioBRIDGE (Wang et al., 2023b) links diverse biological modalities through a knowledge graph, both achieving strong PPI performance. Although BioT5 (Pei et al., 2023) and BioT5+ (Pei et al., 2024) were not explicitly designed for interaction prediction, they still perform competitively on PPI benchmarks. Beyond interaction tasks, multimodal translation is another

emerging direction: MolBind (Xiao et al., 2024b) supports protein-related zero-shot cross-modal retrieval, and BioTranslator (Xu et al., 2023) converts free-text descriptions into biological representations across modalities, enabling more flexible interaction with scientific data.

Collectively, these advances highlight the growing potential of MLLMs to unify heterogeneous protein modalities, enabling more accurate prediction, versatile design, and broader applications in protein science.

## 5 MLLMs for Genomics and Gene

MLLMs and LLMs are rapidly advancing genomics by enabling tasks such as sequence modeling, gene function prediction, functional annotation, and knowledge retrieval. Compared to traditional computational approaches, these models offer greater flexibility, interpretability, and the ability to integrate heterogeneous biological data (Cheng et al., 2024; Hu et al., 2025a; Jin et al., 2024). In this section, we review recent progress from two perspectives. First, we introduce LLMs for genomics, covering their applications in molecular and drug design, functional annotation, gene and variant prioritization, regulatory network modeling, and sequence-level protein or gene tasks. Second, we focus on MLLMs for genomics and gene function prediction, highlighting how multimodal integration of sequences, biological data, and language enables richer reasoning, interpretable predictions, and generalist genomic analysis. Table E3, Table F5, Table F6 and Figure 5 summarize models, datasets, and the research landscape.

### 5.1 LLMs for Genomics

LLMs are rapidly transforming bioinformatics and genomics, with applications spanning molecular and drug design, functional annotation, gene and variant prioritization, regulatory network modeling, sequence analysis, and synthetic data generation (Cheng et al., 2024; Hu et al., 2025a; Chang et al., 2024; Jin et al., 2024; Hou et al., 2025; Toufiq et al., 2023). In molecular design, models such as GexMolGen (Cheng et al., 2024) align gene expression features with chemical structures to enable gene-guided de novo molecule generation. For functional annotation and knowledge retrieval, LLMs are evaluated on summarizing gene sets (Hu et al., 2025a), discovering gene–disease associations (Chang et al., 2024), and augmenting biomedical search with APIs (Jin et al., 2024), while Gene-Turing (Hou et al., 2025) provides systematic benchmarks. In gene and variant prioritization, LLM-based approaches (Toufiq et al., 2023; Liang et al., 2024a; Li et al., 2025d) integrate literature, biological data, and phenotypes to rank causative genes, with automated pipelines supported by API-driven

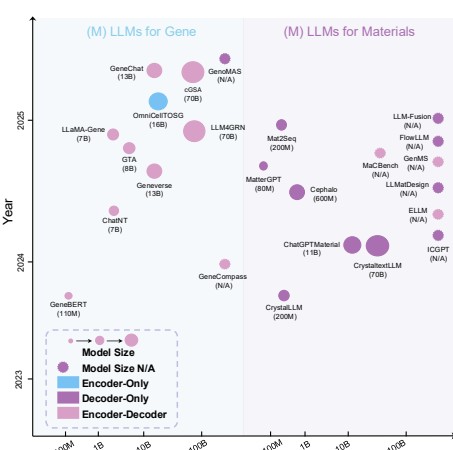

Figure 5: Distribution of MLLMs for gene and materials, presenting each model's release date, scale, and architecture.

workflows (Kim et al., 2024; Khan et al., 2024). For network modeling, LLMs aid cancer driver gene discovery (Zeng et al., 2025) and reconstruct regulatory networks from single-cell and multi-omics data (Wang & Tian, 2025). In sequence-level tasks, models like ProGen (Madani et al., 2023b) generate functional proteins, while others annotate genes and structures directly from sequence data (Duan et al., 2025; Zhu et al., 2022; Liu et al., 2024a; Akotenou & El Allali, 2025; Shmelev et al.). Beyond these, LLMs support antimicrobial resistance prediction (Yoo, 2025), variant effect modeling (Hegde et al., 2025), and even generate synthetic training data for fine-tuning and benchmarking (Majumdar et al., 2024). Together, these studies highlight the broad and transformative role of LLMs in genomics, offering new levels of automation, accuracy, and creativity for precision medicine.

### 5.2 MLLMs for Genomics and Gene Function Prediction

The integration of MLLMs into genomics has introduced a transformative paradigm for gene function prediction, gene expression modeling, and broader biological tasks (Liu et al., 2024e; Dhanasekar et al., 2025; Bhattacharya et al., 2024; Richard et al., 2024; Honig et al., 2024; Mo et al., 2021). Traditional methods based on sequence homology, ontology classification, or narrow

supervised models often lack flexibility and interpretability. In contrast, MLLMs enable free-form reasoning and cross-modal understanding. For example, GeneChat (Dhanasekar et al., 2025) reframes gene function prediction as a language generation task, combining DNABERT-2 (Zhou et al., 2023b) as a gene encoder with Vicuna-13B (Chiang et al., 2023) as a decoder to produce rich natural-language descriptions from raw DNA input. Extending this idea, Geneverse (Liu et al., 2024e) provides a suite of open-source models tailored to genomic and proteomic data, demonstrating strong results in gene/protein function summarization and spatial transcriptomics. ChatNT (Richard et al., 2024), built on the Nucleotide Transformer (Dalla-Torre et al., 2025), supports unified instruction-based inference across DNA, RNA, and protein tasks, making advanced analyses more accessible. Other methods, such as GTA (Honig et al., 2024) and GeneBERT (Mo et al., 2021), further improve regulatory modeling by aligning sequence features with language embeddings or leveraging multimodal pretraining. Despite ongoing challenges—such as limited annotations and multimodal heterogeneity—these advances highlight the potential of MLLMs as generalist, interpretable, and conversational engines for genomics and molecular biology (Bhattacharya et al., 2024).

## 6 MLLMs for Material Science

The use of MLLMs in materials science is still at an early stage but shows strong potential. By integrating text (1D), images (2D), and geometric structural data (3D), these models promise to accelerate material discovery, property prediction, and design optimization (Boyar et al., 2025; Alampara et al., 2024; Buehler, 2024; Pyzer-Knapp et al., 2025). In this section, we review progress from two angles: (1) we discuss LLMs for material discovery, highlighting their role in crystal structure generation, property prediction, and inverse design. (2) we turn to MLLMs for material discovery, where multimodal fusion of textual, visual, and structural representations further enhances property estimation, data extraction, and design pipelines. Table E4 and Figure 5 summarize models and the research landscape.

### 6.1 LLMs for Material Discovery

Recent advancements show that LLMs can significantly aid materials discovery by generating crystal structures, predicting properties, and supporting inverse design (Deb et al., 2024; Antunes et al., 2024; Gruver et al., 2024; Liu et al., 2025b; Jia et al., 2024; Chen et al., 2024b; Yang et al., 2024a; Sriram et al., 2024; Yan et al., 2024; Wang et al., 2024d; Grandi et al., 2025). CrystaLLM (Antunes et al., 2024) autoregressively generates CIF sequences to produce plausible crystal structures. MatterGPT (Chen et al., 2024b) targets properties such as formation energy and band gap and enables multi-property inverse design, demonstrating control over both lattice-insensitive and lattice-sensitive attributes (Chen et al., 2024b). LLMatDesign (Jia et al., 2024) provides an agentic, iterative framework where LLMs propose material modifications, while domain-aware prompt engineering further boosts property prediction (Liu et al., 2025b). FlowLLM (Sriram et al., 2024) couples LLMs with Riemannian Flow Matching to refine representations and generate stable, novel materials. CrystaltextLLM (Gruver et al., 2024) fine-tunes LLMs by encoding atomistic data as text and using energy calculations for stability prediction. Deb et al. (2024) demonstrate ChatGPT's ability to suggest compositions and processing routes, accelerating design. GenMS (Yang et al., 2024a) combines language conditioning with diffusion to generate low-energy crystal structures, and Mat2Seq (Yan et al., 2024) offers SE(3)- and periodic-invariant crystal sequences for robust LM generation. Finally, studies on material selection show that prompt-refined LLMs can assist decisions by comparing expert recommendations (Grandi et al., 2025). Collectively, these advances expand the searchable chemical space and strengthen data-driven materials design.

### 6.2 MLLMs for Material Discovery

The integration of MLLMs into materials science is advancing rapidly for discovery and property prediction (Boyar et al., 2025; Alampara et al., 2024; Buehler, 2024; Pyzer-Knapp et al., 2025). A key direction is multimodal fusion of text, images, and molecular representations; for example, LLM-Fusion (Boyar et al., 2025) flexibly ingests SMILES/SELFIES/fingerprints to enhance property prediction over unimodal baselines. Cephalo (Buehler, 2024) applies vision–language integration to bio-inspired materials, combining images and text from documents and experiments for property estimation and design optimization. MaCBench (Alampara et al., 2024) identifies current limitations—especially spatial reasoning and cross-modal synthesis—highlighting the need for stronger multimodal reasoning. Recent work also targets automatic extraction of materials data from literature and visual content to enable scalable prediction (Pyzer-Knapp et al., 2025). Overall, these

multimodal approaches are poised to transform materials discovery by enabling robust, data-driven design pipelines for both research and industrial applications.

## 7 FUTURE DIRECTIONS

Looking ahead, we identify five practical directions for applying MLLMs in science (with full details in Appendix C). (1) *Diffusion LLMs and MLLMs (dLLMs/dMLLMs) paradigms for scientific workflows.* By replacing left-to-right decoding with iterative mask–denoise refinement, diffusion-based LLMs and MLLMs (Nie et al., 2025; Google DeepMind, 2025) enable parallel generation, stronger global coherence, and natural control over sequence length, layout, and schemas across text, vision, and audio. Combined with lightweight schedulers and confidence-aware decoding, the same mechanisms can also support end-to-end laboratory workflows: they can produce ELN/LIMS-ready procedures, tables, and figures, integrate retrieval, docking, or DFT/MD simulations, and maintain audit trails of sources, constraints, and uncertainties to facilitate plan–execute–revise cycles (Yu et al., 2025a). (2) *Molecular design.* Future models should incorporate basic physical and, where necessary, quantum constraints. They need to move beyond static structures to capture molecular dynamics, while integrating spectroscopy, microscopy, and simulation data in interpretable and uncertainty-aware ways (Lee et al., 2025). (3) *Protein science.* Progress requires moving from single snapshots to ensembles and time-dependent representations. Models should scale to all-atom resolution when necessary, for example through coarse-to-fine decoding and equivariant architectures, and they should embed biophysical constraints during training and inference to ensure plausible structural and functional predictions (Hayes et al., 2025). (4) *Genomics.* Domain-specific encoders must respect reverse-complement symmetry and capture long-range regulatory dependencies. Advances will come from combining sequence modeling with single-cell and spatial transcriptomics, imaging, and clinical text, while grounding outputs in biological ontologies and designing them for clinical use with proper calibration, privacy protection, and provenance tracking (Dhanasekar et al., 2025). (5) *Materials science.* Progress depends on physics-informed priors that encode conservation laws, symmetry, and periodicity, together with graph/3D representations that link composition and structure to properties. Coupling models to multiscale dynamics or to fast surrogates of DFT/MD helps predict path-dependent behavior (e.g., phases, defects) and accelerates both screening and inverse design (Antunes et al., 2024).

## 8 CONCLUSION

This work provides a comprehensive overview of recent advances in MLLMs for science, highlighting representative architectures, datasets, and benchmarks, as well as their emerging applications in science. Beyond cataloging progress, we also emphasize the growing role of diffusion-based LLMs in multimodal generation and reasoning. Looking ahead, MLLMs hold the potential to reshape the way scientists explore and integrate diverse data sources. Continued progress will require addressing open challenges in factual reliability, modality-specific reasoning, interpretability, and ethical deployment. By synthesizing current advances and pointing toward future directions, this work aims to serve as both a reference and a foundation for further research in multimodal scientific AI.

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

# A GENERAL OVERVIEW FOR LLMS AND MLLMS

In this section, we aim to provide readers with a coherent background framework by reviewing the foundational components and architectural innovations of LLMs and their multimodal counterparts (MLLMs). By systematically discussing their core components, training paradigms, multi-modal extensions, we establish a clear understanding of how these models function. We also present a high-level overview of the framework for the LLMs and MLLMs in Figure 6. This overview sets the stage for the the main paper, where we turn to the specific applications of MLLMs in scientific domains.

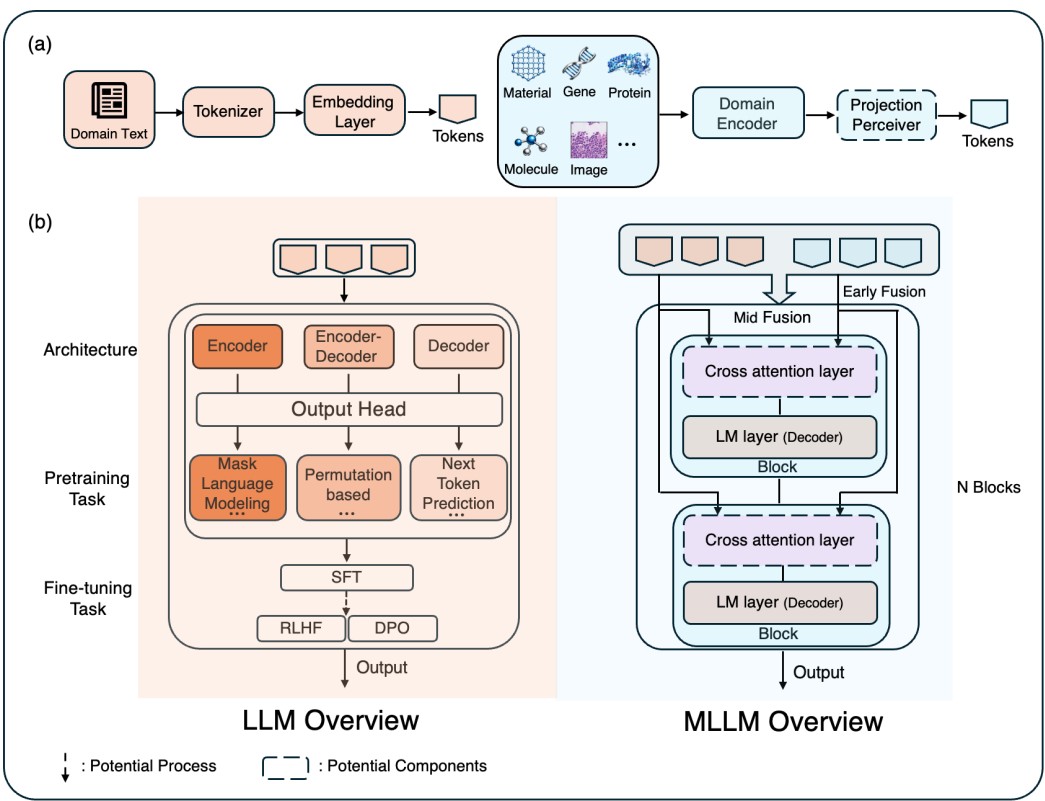

Figure 6: The overview of the architecture for LLMs and MLLMs. The figure illustrates three major LLM paradigms (encoder-only, encoder-decoder, and decoder-only) with their pretraining and fine-tuning tasks(IT means Instruction tuning, and RLHF means Reinforcement Learning from Human Feedback). LLMs serve as the foundation of MLLMs. The latter integrate modality-specific encoders to extract representations from diverse data modalities. These representations are then projected or injected into the language embedding space via projection layers or perceivers, followed by fusion of multi-modal embeddings to generate the final output.

**Core Components of LLMs.** The backbone of modern LLMs is the Transformer architecture (Vaswani et al., 2017), which revolutionized natural language processing by introducing self-attention mechanisms. At the input stage, text is first processed into tokens through a tokenizer. Depending on the domain, these tokens may correspond to words, subwords, or characters, while specialized tokenizers are designed for structured domains such as DNA sequences or chemical molecules. Each token is then mapped into a dense vector representation by the embedding layer, where positional embeddings (absolute or relative type)inject sequence order information into the otherwise permutation-invariant architecture. The central component of LLMs consists of stacked Transformer blocks. Based on the original Transformer architecture, three mainstream LLM architectures have emerged: encoder-only, represented by the BERT (et al., 2018) family; decoder-only, exemplified by LLaMA (Liang, 2024a); and encoder-decoder, represented by models such as

GLM (Du et al., 2021). Specifically, each block(often referred to as an LM layer) contains multi-head self-attention layers, feed-forward networks, normalization steps, and residual connections, which together enable the model to capture long-range dependencies across large contexts. Finally, the model is equipped with an output layer: generative models project hidden representations to vocabulary probabilities, while encoder-based models connect to task-specific heads for classification, retrieval, or regression. These components collectively determine the expressive power and adaptability of LLMs across tasks.

**Training Objectives and Techniques.** The objectives used in training LLMs directly shape their behavior and suitability for downstream tasks. Autoregressive models, exemplified by the GPT family (Radford et al., 2019), learn to predict the next token in a sequence, which makes them particularly effective for text generation. In contrast, masked language modeling (MLM), popularized by BERT (Devlin et al., 2019), involves randomly masking tokens and training the model to recover them, producing strong bidirectional representations useful for understanding tasks. Other approaches, such as XLNet (Yang et al., 2019), introduce permutation-based objectives to combine the strengths of both autoregressive and masked methods. Beyond these pretraining objectives, finetuning strategies are used for models to better perform on downstream tasks or align better with human preferences. alignment with human preferences has become increasingly important. By training LLMs on a dataset consisting of instruction and output pairs or using reinforcement learning with human feedback, instruction tuning bridges the gap between the next-word prediction objective and users' objective of having LLMs adhere to human instructions (Shengyu et al., 2023; Ouyang et al., 2022). These techniques have been critical to the deployment of interactive models like ChatGPT and GPT-4.

**Multimodal Large Language Models (MLLMs).** While LLMs excel in language tasks, many real-world applications demand reasoning across multiple modalities such as text, images, audio, or structured scientific data. MLLMs extend LLMs by introducing architectures capable of integrating heterogeneous inputs. Typically, they first leverage modality-specific encoders which are aligned with the text modality via contrastive learning to transform non-textual modalities into language-aligned embeddings , such as pretrained CLIP visual encoder (Li et al., 2023a). Textual inputs are processed in a manner similar to LLMs. These embeddings may be then projected into the language space through a projection layer or a perceiver module,followed by the adoption of various fusion strategies to integrate information across modalities. Early-fusion approaches combine embeddings from different modalities at the input stage, often through direct concatenation (Zhu et al., 2023). In contrast, late-fusion architectures encode each modality independently and combine their outputs only at the reasoning or decision stage. The strategy has become less common as LLM capabilities have advanced. More sophisticated Fusion strategy can occur in the mid stage. for example, cross-attention architectures allow one modality to attend to features from another, exemplified by models such as Flamingo (Alayrac et al., 2022) and BLIP-2 (Li et al., 2023c), which achieve strong results in vision-language tasks. To address the prohibitive cost of retraining entire LLMs for multimodal tasks, adapter-based techniques such as LoRA (Hu et al., 2022) introduce lightweight, trainable components into frozen models. These advances make MLLMs more efficient and practical for specialized multimodal scenarios.

**Pretraining Datasets and Modalities.** The performance of LLMs and MLLMs is intimately tied to the scale and diversity of their pretraining datasets. For text, models typically rely on large and diverse corpora such as Wikipedia, Common Crawl, PubMed, and patent databases. In the multimodal domain, paired datasets such as LAION-5B (Schuhmann et al., 2022) provide billions of image-text pairs for training vision-language systems. Scientific and technical applications require more specialized resources. Biological sequence data (e.g., UniProt), molecular graphs (e.g., ChEMBL), and crystallographic structures are increasingly integrated into pretraining. Moreover, structured ontologies and knowledge graphs such as the Gene Ontology (GO) or UMLS are used to augment factual reasoning and reduce hallucinations. The combination of unstructured and structured data creates rich environments for pretraining models capable of bridging multiple domains.

**Common Use Cases Across Domains.** The versatility of LLMs and MLLMs is reflected in their broad range of use cases. One major paradigm is zero- or few-shot inference, where models solve novel tasks with little to no labeled data by leveraging their pretraining knowledge. When higher domain specificity is needed, fine-tuning can adapt general-purpose LLMs to specialized applications such as drug discovery, clinical prediction, or materials design. Increasingly, LLMs are being used as tool-augmented systems. By integrating with external APIs, databases, or scientific engines

such as AlphaFold DB, models can dynamically expand their capabilities beyond what is encoded in their parameters. A further evolution of this idea is the emergence of agent-based workflows, where models orchestrate multi-step reasoning, execute code, and autonomously coordinate experiments or data analysis pipelines.

# B  MLLMs BRIDGING MOLECULAR SCIENCE AND BIOMEDICINE

The biomedical field encompasses a vast array of disciplines, from fundamental biological research to complex clinical applications (Wang et al., 2024a), and naturally involves a variety of data modalities, amog which analyses of molecules, proteins, genes, and cells play a crucial role. MLLMs have opened new possibilities for integrating heterogeneous biomedical data, enabling not only multi-molecular data fusion (Liu et al., 2024e; Liang, 2024b) but also the combination of microscopic-level data(e.g., molecular or cellular information) with macroscopic-level data such as pathology images (Lin et al., 2024; Xu et al., 2025), offering valuable insights into disease machanisms and improving diagnostic accuracy. In this section, we primarily focus on the recent surge of studies employing MLLMs to integrate molecular science with biomedicine,along with their methodological approaches. Table E5 summarizes the models discussed in this section. Based on existing advancements, we discuss the limitations identified and outline future directions for further integrating molecular science into biomedicine.

## B.1  LLMs FOR BIOMEDICINE

Genomic, epigenetic, and transcriptomic analyses such as gene pathway finding, gene expression analysis, and so on, greatly facilitate our understanding of biological processes and mechanisms in both normal organism development and disease (Wang et al., 2025e). These sequences modalities are escpecially suitable for LLMs to process. Some methods (Wang et al., 2025e; Afonja et al., 2024) integrates domain knowledge and study context into LLMs to enable gene analysis at different levels of granularity. Specifically, (Wang et al., 2025e) focuses on gene set enrichment analysis to explicitly consider gene interactions and regulatory relationships within gene sets, while (Afonja et al., 2024) aims to infer gene regulatory networks (GRNs). Together, these approaches facilitate the characterization of caner-related pathways and the elucidation of disease mechanisms, ultimately aiding the idendification of effective treatments. In more recent applications, GenoMAS (Liu et al., 2025a) orchestrating six specialized LLM agents, each contributing complementary strengths to a shared analytic canvas, is applied to gene expression analysis which exposes biologically plausible gene-phenotype associations corroborated by the literature.

## B.2  MLLMs FOR CROSS MODAL TASKS

With the advent of MLLMs, it has become possible to analyze biomedical problems from multiple perspectives — not only at the macroscopic level (e.g., images and audio) but also at the molecular level. Unlike traditional multimodal fusion approaches (Schouten et al., 2025; Chaabene et al., 2025; Mu et al., 2020), which rely on human-designed summarization, MLLMs can autonomously provide highly interpretable insights and handle cross-modal tasks such as visual question answering and report generation.

(1) *Multi-omics Fusion Models.* Combining omics data into biomedical research has achieved some success (Duan et al., 2021). Current research primarily focuses on developing methods to effectively harmonize diverse omics modalities (Ye & Tang, 2025). One line of research leverages the intrinsic capability of MLLMs to directly fuse heterogeneous omics data, such as genes, molecules, and proteins. Geneverse (Liu et al., 2024e) fine-tunes LLaVA by incorporating protein structural information, gene expression profiles, and functional descriptions as inputs. BioMedGPT (Luo et al., 2023a) further integrates a broader range of biomedical modalities with different encoders, unifies the feature spaces of molecules, proteins, and natural language through encoding and alignment. Another line of research first transforms different modalities into a shared representation before feeding them into MLLMs. LLaMA-Gene (Liang, 2024a) trains a single BPE (Byte Pair Encoding) tokenizer to encode genes, proteins, and natural language sequences without additional markers and further converts gene-related task data into a unified format for instruction fine-tuning, constructing a unified model for diverse gene tasks. Collectively, these works support downstream applications such as protein identification and marker gene discovery with the potential to greatly accelerate the discovery of new drugs and therapeutic targets.

(1) *Richer Multimodal Fusion in Biomedicine.* At the same time, beyond exploring modality fusion within a specific domain or dimension, there have been growing efforts to integrate a broader range of modalities. For example, multi-omics data are fused with cell even organ type data, of-

fering more subtle information about the condition. OmniCellTOSG (Zhang et al., 2025a) encodes textual annotations with an LLM and leverages a graph neural network (GNN) to capture the topology of signaling(TOSG) networks labeled with annotations like organ, cell subtype, and quantitative gene and protein data. By integrating these two representations, it constructs patient-specific single-cell TOSG maps, thereby enabling precise cell classification, cancer cell state prediction, and other clinically relevant tasks transforming research in life sciences, healthcare, and precision medicine. SpaLLM (Li et al., 2025c) combines LLM representations from single-cell transcriptomics with spatially resolved multi-omics data (e.g., RNA, chromatin accessibility, proteins), enabling precise identification of functionally specialized cell types, providing essential molecular and spatial references for disease diagnosis. Recently, another popular direction in MLLM-based research has been to leverage spatial transcriptomics (ST) technologies, which provide both molecular signatures and the spatial localization of cells within tissues. ST-ALign (Lin et al., 2024) leverages ST technology to achieve fine-grained alignment between histological morphology and molecular features, including image–gene alignment at both the spot and niche levels, following by an Attention-Based Fusion Network used to fuse visual and genetic features. Extending spatial transcriptomics to pathology, mSTAR and spEMO (Xu et al., 2025; Liu et al., 2025e) integrate microscopic slides, macroscopic reports, and gene expression via multi-level alignment into a pathology foundation model, enabling tasks such as diagnosis, molecule prediction, survival analysis, and report generation. Furthermore, spEMO introduces the novel task of multimodal alignment, offering a new perspective to evaluate information retrieval ability and guide the development of future pathology foundation models.

### B.3 OUTLOOK

Although MLLMs have begun to explore the integration of multiple modalities, current progress remains at an early stage. For instance, while some models (Li et al., 2025c; Liu et al., 2024e; Liang, 2024a) have been trained on multi-omics data simultaneously, few are capable of jointly processing image-based data, largely due to the weak consistency across such heterogeneous modalities. integrating more diverse data types thus remains challenging. A few models, such as (?), have attempted to combine pathological images with genomic information for disease diagnosis, but such approaches are still limited. There remains a clear need for more comprehensive methods that effectively integrate diverse multimodal data in the future. A promising direction for sustainable progress is to curate large-scale, comprehensive multimodal benchmarks and datasets to facilitate the development of future methods.

## C   EMERGING HOT TOPICS AND FUTURE DIRECTIONS

In this section, we (1) examine several *emerging hot topics*, with a particular focus on diffusion-based paradigms that are reshaping large language models and their multimodal extensions, and (2) discuss *future directions* in scientific applications of MLLMs, covering domain-specific challenges and opportunities across molecular science, protein modeling, materials discovery, and genomics.

### C.1   EMERGING HOT TOPICS

The rapid progress of large language models has spurred a new wave of research into alternative training and decoding paradigms, as well as extensions to multimodal understanding and generation. In this section, we highlight two directions that have recently gained considerable momentum. The first is diffusion large language models (dLLMs), which replace the conventional autoregressive decoding strategy with an iterative mask–denoise process and have shown promising advances in reasoning, controllability, and efficiency. The second is diffusion multimodal large language models (dMLLMs), which extend this paradigm to vision, audio, and other modalities, enabling more flexible cross-modal reasoning and structured generation. Together, these emerging topics illustrate how diffusion-based methods are shaping the future landscape of language and multimodal modeling.

#### C.1.1   DIFFUSION LARGE LANGUAGE MODELS

dLLMs replace the traditional left-to-right next-token prediction paradigm with a mask-and-denoise process over discrete tokens. Instead of generating text sequentially with unidirectional attention, dLLMs begin from a heavily masked (or absorbed) sequence and iteratively denoise it using bidirectional attention. This design enables parallel decoding of many tokens at once, providing explicit trade-offs between quality, latency, and controllability through adjustable steps and scheduling (Yu et al., 2025a; Gong et al., 2022; Zheng et al., 2023; Song et al., 2025; Liu et al., 2025f). Compared with autoregressive (AR) models, which suffer from rigidity in mid-sequence editing and lack global structural control, diffusion-based decoding offers greater flexibility and coherence.

(1) *Core Mechanics.* The forward process in dLLMs typically applies random masking or absorbing states, while the reverse process learns to reconstruct clean tokens from noisy inputs. Recent advances, such as reparameterized discrete diffusion (RDM), reduce training variance and enable confidence-aware decoding by prioritizing high-confidence tokens during generation (Zheng et al., 2023). Training objectives span from NLL-equivalent token prediction to reweighting strategies at the token or sequence level. For example, multi-granularity diffusion (MGDM) emphasizes difficult tokens and subgoals to enhance complex reasoning (Ye et al., 2024a). At inference, specialized schedulers such as dilated unmasking explicitly minimize conditional entropy in each round, thereby reducing the number of iterations (Luxembourg et al., 2025).

(2) *Scaling Strategies.* Two main approaches have emerged for scaling dLLMs. The first is training from scratch, exemplified by LLaDA, which pre-trains an 8B-parameter diffusion LLM on 2.3T tokens and demonstrates competitive or superior performance to comparable AR baselines, particularly on reversal-style tasks that reveal AR brittleness (Nie et al., 2025). The second strategy adapts pretrained AR models by gradually relaxing the causal mask and shifting prediction targets, yielding variants such as DiffuGPT & DiffuLLaMA that achieve strong zero/few-shot and fill-in-the-middle abilities with significantly reduced training cost (Gong et al., 2024).

(3) *Capabilities and Directions.* Diffusion decoding has opened new research avenues across multiple fronts: *(i) Reasoning and planning.* Diffusion-of-Thought supports parallelized chain-of-thought and multi-step self-correction (Ye et al., 2024b), while MGDM reports substantial improvements on tasks such as Countdown, Sudoku, and SAT (Ye et al., 2024a). Recent work like d1 combines supervised fine-tuning with a diffusion-compatible policy-gradient method (diffu-GRPO), further improving math, logic, and coding performance (Zhao et al., 2025). *(ii) Program synthesis and structured generation.* DiffuCoder introduces analysis tools for "AR-ness" of dLLMs and a coupled-GRPO RL procedure, matching or beating similar-sized AR coders on several leaderboards (Gong et al., 2025). For controllable outputs (JSON/tables), the S3 scaffolding method uses schema templates and null tokens to achieve high structural validity without retraining (Xiong et al., 2025). *(iii) Seq2Seq and one-step generation.* DiffuSeq extends diffusion to conditional text generation (Gong

et al., 2022). DLM-One distills iterative denoising into a single forward pass via score-based distillation—reporting up to 500× speedups on classic Seq2Seq tasks at near-teacher quality (Chen et al., 2025a). *(iv) Systems & efficiency.* At inference, dilated unmasking reduces rounds from $O(B)$ to roughly $O(\log B)$ per block (Luxembourg et al., 2025); Fast-dLLM adds block-wise KV caching plus confidence-gated parallel decoding, reporting up to 27.6× speedups with minimal accuracy loss (Wu et al., 2025). Block diffusion interleaves AR across blocks with diffusion within blocks, closing perplexity gaps while preserving parallelism (Arriola et al., 2025). *(v) Industrial interest.* Google DeepMind's Gemini Diffusion signals growing product-level exploration of text diffusion (Google DeepMind, 2025).

(4) *Safety Outlook.* The novel dynamics of dLLMs introduce distinct safety challenges. Parallel decoding and mask-aware mechanisms create new attack surfaces, and recent jailbreak methods such as PAD and DIJA achieve high success rates across multiple diffusion models (Zhang et al., 2025c; Wen et al., 2025). These results suggest that AR-based defenses cannot be directly applied, underscoring the need for diffusion-native alignment and guardrails.

(5) *Takeaway.* dLLMs combine parallelism, global coherence, and fine-grained controllability, positioning them as a promising alternative—and in some regimes, a superior paradigm—to autoregressive models (Yu et al., 2025a). With both training-from-scratch and AR-adaptation paths maturing, and with rapidly improving inference-time efficiency, dLLMs are evolving from niche prototypes to competitive large-scale systems.

(6) *Open Problems and Future Directions.* Key challenges remain: (i) establishing theoretical foundations for scheduling, convergence, and optimality; (ii) developing scalable diffusion-native alignment and RLHF methods (Zhao et al., 2025); (iii) hybridizing diffusion with AR, retrieval, and external tools (Arriola et al., 2025; Ye et al., 2024a); (iv) designing standardized evaluation protocols for latency–quality trade-offs and structural validity; (v) advancing security via mask-aware defenses and robust red-teaming (Zhang et al., 2025c; Wen et al., 2025); and (vi) optimizing serving systems for KV-cache consistency, adaptive decoding, and distributed/edge deployment (Wu et al., 2025; Luxembourg et al., 2025).

### C.1.2 Diffusion Multi-modal Large Language Models.

dMLLMs are also attracting increasing attention in the multimodal domain. Compared to autoregressive approaches, iterative mask–denoise refinement provides *global context modeling*, *parallel token prediction*, and natural support for structure priors (e.g., layouts, JSON schemas) as well as fill-in-the-middle editing. These properties make diffusion particularly suitable for vision–language, audio–language, and other structured multimodal tasks, while offering explicit quality–latency trade-offs through the choice of denoising steps (Yu et al., 2025a).

(1) *Representative Models.* Several recent systems demonstrate the potential of diffusion in multimodal scenarios. (i) *Vision–language.* Llada-v extends LLaDA with visual instruction tuning while retaining diffusion-style parallel decoding, enabling visual question answering and multimodal dialogue (You et al., 2025). Dimple adopts a two-stage training paradigm: an initial AR phase aligns vision and text representations and supports instruction following, after which diffusion decoding is reinstated to recover parallelism and structural control. At inference, Dimple incorporates confident decoding and explicit structure priors (e.g., JSON length control), achieving state-of-the-art results with fewer denoising steps (often less than one-third of the response length) (Yu et al., 2025b). (ii) *Audio–language.* DIFFA freezes Whisper and a diffusion LLM backbone, training only lightweight dual adapters (semantic and acoustic). This adapter-based design yields strong performance across multiple audio–language benchmarks at modest data and compute cost, highlighting the efficiency of multimodal diffusion tuning (Zhou et al., 2025b). (iii) *Broader ecosystem.* Beyond academic prototypes, Gemini Diffusion illustrates early integration of diffusion-style generation into large-scale product pipelines, signaling practical interest in retrieval- and tool-augmented multimodal agents (Google DeepMind, 2025).

(2) *Capabilities and Engineering Patterns.* Diffusion multimodal models inherit many of the strengths of their text-only counterparts. (i) *Controllability and structure.* By conditioning on scaffolds such as schemas or layouts, these models substantially reduce format errors and hallucination in chart/table reasoning and structured generation; S3-style prompting can be readily reused in multimodal contexts (Yu et al., 2025b; Xiong et al., 2025). (ii) *Throughput and latency.* Inference

accelerations developed for dLLMs, including KV-cache reuse, confidence-gated parallel decoding, and dilated scheduling, transfer cleanly to vision and audio modalities (Wu et al., 2025; Luxembourg et al., 2025). (iv) *Applications.* Iterative refinement proves beneficial for fact-faithful summarization (Arg-LLaDA) and for constrained scientific design/optimization where diffusion acts as a constrained sampler over feasible manifolds (Li et al., 2025a; Kong et al., 2024). Other applications include controllable user-facing content generation such as poll/question generation with attribute control (Cheng & Li, 2024).

(3) *Risks and Challenges.* Despite these advances, several challenges remain open. (i) *Security.* Mask-aware, parallel denoising can amplify multimodal jailbreak attacks, including cross-modal prompt mixing and masked injection; diffusion-native safeguards are still underdeveloped (Zhang et al., 2025c; Wen et al., 2025). (ii) *Long-context efficiency.* Processing long videos or extended speech raises issues of memory and cache consistency across denoising steps, requiring more principled architectural solutions (Wu et al., 2025; Luxembourg et al., 2025). (iii) *Data and alignment.* High-quality multimodal instruction data remain scarce; balancing frozen-backbone adapters (e.g., DIFFA) with full-parameter training (e.g., Dimple) is still an open question for efficient scaling (Zhou et al., 2025b; Yu et al., 2025b).

(4) *Future Directions.* Promising research avenues include: (i) designing unified diffusion agents that couple vision, audio, and text with retrieval and tool use; (ii) developing verifiable generation under hard structure/layout constraints; (iii) scalable alignment via multimodal preference modeling and reinforcement learning for diffusion; (iv) building diffusion-native defenses and safety benchmarks; and (v) systems co-design for efficient step-adaptive serving, block-wise diffusion, and distributed or edge inference (Arriola et al., 2025; Xiong et al., 2025; Wu et al., 2025; Luxembourg et al., 2025).

## C.2 FUTURE DIRECTIONS

MLLMs have profoundly transformed the research landscape across domains including molecular science, protein science, material discovery, genomics, medicine, and beyond (Luo et al., 2023a; Liu et al., 2025d; Dhanasekar et al., 2025; Boyar et al., 2025). Despite these advances, there remain substantial gaps between the current state of the art and the long-term vision of autonomous, trustworthy, and general-purpose scientific agents. To bridge this gap, we identify future directions that can be broadly categorized into domain-specific challenges and cross-disciplinary opportunities, with the goal of guiding research toward impactful advances.

### C.2.1 MLLMs FOR MOLECULAR DESIGN.

Molecular design demands models that can faithfully capture the geometry, dynamics, and physical constraints of molecules. At this juncture, we identify several promising research avenues that merit particular attention. (1) *Physical-constraint modeling.* Current MLLMs primarily rely on sequence- or graph-based representations, but often fail to enforce fundamental physical constraints such as atomic distance limits, bond angles, or quantum-level properties. Embedding such priors into the modeling pipeline can significantly improve robustness and interpretability. (2) *Modeling dynamics.* Most existing approaches treat molecules as static entities, whereas real-world properties depend heavily on dynamic behavior. Extending MLLMs to incorporate temporal molecular dynamics would open new opportunities in reaction prediction, drug discovery, and material synthesis. (3) *Complex data integration.* Molecular research spans diverse modalities, including spectroscopy, microscopy, and quantum simulation data. Designing models capable of integrating such heterogeneous data while respecting inter-modality constraints (e.g., protein–ligand interactions) is a key challenge. (4) *Quantum-aware representations.* A promising direction is to develop encoders grounded in quantum chemistry and physics, moving beyond atomistic descriptors toward foundation models that operate directly at the quantum level.

### C.2.2 MLLMs FOR PROTEIN SCIENCE

Proteins present distinctive challenges for MLLMs owing to their rugged, high-dimensional conformational landscapes and the tight coupling between structure, dynamics, and function. Progress in this area will likely hinge on advances along three fronts: (1) *Protein dynamics.* Most current LLM-based approaches operate on static snapshots (e.g., single structures or sequences), whereas

many biological functions are mediated by ensembles, transitions, and rare events. Incorporating temporal information—through trajectory-aware representations, coarse-to-fine dynamical priors, or learned surrogates of molecular simulation—remains underexplored yet essential for capturing allostery, binding pathways, and conformational selection. (2) *All-atom modeling.* To achieve biochemical fidelity, models must scale beyond residue- or coarse-grained abstractions toward all-atom resolution when warranted. This entails addressing substantial challenges in data volume and quality, long-range interactions, and computational cost. Promising directions include hybrid granularity (coarse-to-fine decoding), equivariant architectures, and teacher–student distillation from physics-based engines to amortize expensive detail into lightweight predictors. (3) *Physical priors.* Ensuring physical plausibility requires embedding biophysical constraints into both learning and inference. Constraints such as steric exclusion, hydrogen bonding patterns, rotamer preferences, electrostatics, and solvation effects can be introduced via energy-inspired regularization, constraint-aware decoding, or differentiable scoring functions. Such priors improve sample quality, stabilize training, and facilitate interpretation of model hypotheses.

### C.2.3 MLLMs for Material Science

Materials science is inherently multiscale: atomic arrangements and compositional motifs give rise to mesoscale structures and ultimately emergent macroscopic properties. This hierarchy creates both challenges and opportunities for MLLMs. We outline three research directions that, in our view, are especially promising: (1) *Embedding physical priors.* Robust generalization in materials requires models that respect conservation laws, crystallographic symmetries, and periodic boundary conditions. Incorporating such priors can be achieved via symmetry-/equivariance-aware architectures (e.g., $SE(3)$- or space-group–equivariant layers), periodic convolutions or attention with fractional translations, and energy-/constraint-informed objectives that penalize unphysical predictions. Physics-informed learning not only improves accuracy and sample efficiency but also enhances interpretability and reliability for downstream design. (2) *Graph and 3D-aware encodings.* Faithful structure–property learning hinges on representations that capture local coordination, long-range interactions, and periodicity. Promising approaches include crystal graphs with edge features for bond topology and lattice geometry, voxelized or point-cloud 3D tensors coupled with SE(3)-equivariant networks, and hybrid representations that combine composition-aware language tokens with geometric encoders. For polycrystalline or amorphous systems, hierarchical encodings that bridge atomic neighborhoods to microstructural descriptors (e.g., grains, phases, defects) are critical. (3) *Modeling material dynamics.* Many target properties (e.g., conductivity, elasticity, phase stability) are path- and state-dependent. Integrating molecular/mesoscale dynamics with MLLMs—via differentiable simulators, learned surrogates of MD/DFT, or sequence-of-states generation with uncertainty calibration—can enable predictive modeling of time-dependent behavior and rare events. Coarse-to-fine multiscale schemes (linking atomic MD to continuum models) and step-adaptive inference further reduce cost while retaining fidelity.

### C.2.4 MLLMs for Genomics and Gene Modeling

Genomic modeling with LLMs remains nascent, yet it holds substantial promise for both biomedical research and clinical translation. We highlight six directions that, in our view, are especially consequential: (1) *Domain-specific architectures.* Genomic sequences obey grammars distinct from natural language (e.g., reverse-complement symmetry, motif locality, long-range regulatory dependencies). Dedicated encoders—such as k-mer or PWM-based tokenization, reverse-complement–aware embeddings, and DNABERT-style pretraining—should be scaled with explicit inductive biases for strand orientation, periodicity, and promoter/enhancer motif composition. Long-context modeling (chromatin-scale windows) and equivariant or positionally robust attention schemes are likely prerequisites for capturing distal regulation. (2) *Precision medicine.* Clinically useful systems must generalize to rare variants and patient-specific contexts while quantifying uncertainty. Promising approaches include: (i) variant-centric pretraining with functional assays and curated pathogenicity labels; (ii) multi-omics conditioning (genome, transcriptome, epigenome, proteome) with cohort-level normalization; and (iii) calibration- and causality-aware objectives (counterfactual augmentation, conformal prediction) to support safe decision-making and evidence grading. (3) *Multimodal reasoning.* Many phenotypes emerge from interactions between sequence, expression, imaging, and clinical narratives. MLLMs that fuse DNA/RNA with single-cell profiles, spatial transcriptomics, radiology/pathology images, and EHR text require alignment objectives across modalities

(contrastive or cycle-consistent learning), privacy-preserving training (federated or DP-SGD), and representations that remain stable across batches, platforms, and tissues. Such models could enable end-to-end gene–phenotype mapping and mechanism-aware hypothesis generation. (4) *Ontology-grounded learning.* Embedding structured biological knowledge—e.g., Gene Ontology (GO) and Human Phenotype Ontology (HPO)—into pretraining and inference can improve interpretability and biological fidelity. Practical instantiations include knowledge-graph–regularized objectives, constraint-aware decoding that enforces ontology consistency, and retrieval-augmented generation over curated databases to reduce hallucinations and promote traceable evidence. (5) *Clinical deployment.* Translation to practice demands robust interfaces and governance. Key components are validated APIs that interoperate with established resources (e.g., Ensembl, ClinVar), auditable provenance and versioning, shift detection and post-deployment monitoring, and standardized reporting of model confidence and limitations. Attention to data governance, consent, and reproducibility is essential for regulatory acceptance and safe adoption. (6) *3D genome modeling.* Gene regulation depends on 3D chromatin organization (loops, TADs, compartments). Moving beyond linear sequence requires integrating Hi-C/Micro-C and imaging-derived contact maps via geometric encoders (graph transformers with chromatin contacts, SE(3)-aware models) or discrete "3D structure tokens". Joint sequence–structure pretraining with constraint-aware objectives (e.g., enforcing topological consistency) may unlock more accurate prediction of enhancer–promoter interactions and context-specific expression.

### C.2.5 KEY OPPORTUNITIES OF dLLMs AND dMLLMs FOR SCIENTIFIC DISCOVERY

Diffusion models can fill many tokens in parallel, keep the whole output consistent, and follow templates or rules. Multimodal diffusion extends this to images, spectra, micrographs, 3D structures, and time series. In molecules/drug discovery, proteins, genomics, and materials, this leads to the following concrete wins: (1) *Structured outputs you can use immediately.* With mask–denoise decoding and JSON/table templates, the model can produce ELN/LIMS-ready content: steps with timestamps and units, property tables with ranges and confidence, and provenance fields. If you change a solvent or temperature, a quick refinement updates stoichiometry and safety notes without breaking the rest. (2) *Design that respects hard scientific rules.* Encode required constraints (e.g., valence/sterics, space groups and site occupancy, rotamers and clashes) as scaffolds. Each denoising round proposes candidates; fast scorers or small simulators (QSAR, DFT, MD, energy terms) accept/reject and feed back. You get a ranked set of synthesizable molecules, stable crystal prototypes, or robust protein variants. (3) *Plan–execute–revise instead of one-shot generation.* Parallel chain-of-thought drafts multiple synthesis routes or assay protocols at once. Confidence-aware unmasking keeps strong steps and rewrites weak ones. The system can insert checks (yield, hazard class, cost) and suggest plan B/C with different reagents or instruments so labs can pick what fits their resources and risk. (4) *Tight loops with retrieval and domain tools.* At each diffusion step, call literature/patent search, databases, and tools (reaction predictors, DFT/MD, docking). Write the numbers back—conditions, peaks/bands, formation energies—then refine once more to keep text, tables, and figures consistent. This helps gene–function summaries, materials reports, and chemistry writeups line up with evidence. (5) *Handles long and streaming data.* Block-wise or step-adaptive diffusion can summarize microscopy videos, time-lapse experiments, or audio lab logs as they arrive. It flags anomalies (phase change, crack start, contamination) with timestamps and follow-up suggestions, and maintains a running, unit-checked report for shift handover. (6) *Built-in safety and an audit trail.* Before unmasking sensitive content, apply mask rules (e.g., banned reagents or protocols), schedule randomization, and uncertainty gates. Every run records sources used, constraints triggered, and candidates rejected, creating a clear, reproducible record for compliance and peer review.

# D SELECTED BENCHMARKING EVALUATION

## D.1 MOLECULAR PROPERTY PREDICTION

**Experiment setting.** We evaluate on the MoleculeNet benchmark (Wu et al., 2018), which comprises three single-modal binary classification datasets for assessing the expressiveness of pretrained molecular representation methods. Performance is reported as the area under the receiver operating characteristic curve (AUROC).

Table D1: ROC-AUC (%) results on molecular property prediction tasks (BACE, BBBP, HIV) from the MoleculeNet benchmark (Wu et al., 2018). For non-MLLM models, we adopt the results reported in the InstructMol paper (Cao et al., 2023).

| Method | BACE↑ 1513 | BBBP↑ 2039 | HIV↑ 41127 |
|---|---|---|---|
| *Specialist Models* | | | |
| ChemBERTa v2 | 73.5 | 69.8 | 79.3 |
| DMP(TF+GNN) | 89.4 | 77.8 | 81.4 |
| KV-PLM | 78.5 | 70.5 | 71.8 |
| GraphCL | 75.3 | 69.7 | 78.5 |
| GraphMVP-C | 81.2 | 72.4 | 77.0 |
| MoMu | 76.7 | 70.5 | 75.9 |
| MolFM | 83.9 | 72.9 | 78.8 |
| Uni-Mol | 85.7 | 72.9 | 80.8 |
| *LLM Based Generalist Models* | | | |
| Galactica-6.7B | 58.4 | 53.5 | 72.2 |
| Vicuna-v1.5-13b-16k (4-shot) | 49.2 | 52.7 | 50.5 |
| Vicuna-v1.3-7B* | 68.3 | 60.1 | 58.1 |
| LLaMA-2-7B-chat* | 74.8 | 65.6 | 62.3 |
| MolCA(1D) | 79.3 | 70.8 | – |
| MolCA(1D + 2D) | 79.8 | 70.0 | – |
| Instruct-G | 84.3 (±0.6) | 68.6 (±0.3) | 74.0 (±0.1) |
| Instruct-GS | 82.1 (±0.1) | 72.4 (±0.3) | 68.9 (±0.3) |
| MoleculeSTM (Graph) | 80.77 (±1.34) | 69.98 (±0.52) | 76.93 (±1.84) |
| MoleculeSTM (Smiles) | 81.99 (±0.41) | 70.75 (±1.90) | 76.23 (±0.80) |
| Token-Mol (averaged across five runs) | 89.52 (±1.32) | 91.67 (±0.98) | 82.40 (±0.17) |

**Benchmarking Models.** We identify several MLLMs, including InstructMol (Cao et al., 2023), MoleculeSTM (Graph) (Liu et al., 2023), MoleculeSTM (Smiles) (Liu et al., 2023), GIT-Mol (Liu et al., 2024c), Token-Mol Wang et al. (2025c), and M3LLM (Hu et al., 2024), which target the downstream task of molecular property prediction. For non-MLLM models, we adopt the results reported in the InstructMol paper (Cao et al., 2023). Since the model weights of InstructMol, M3LLM, and GIT-Mol are not publicly available, we rely on the reported results of InstructMol from the original paper, while M3LLM and GIT-Mol are excluded from our evaluation. For the remaining models, we rerun the experiments ourselves.

**Analysis.** Overall, as show in Table D1, the results show that MLLM-based models achieve competitive performance in molecular property prediction, but they generally lag behind strong specialist models such as Uni-Mol and MolFM. Among the evaluated MLLMs, Token-Mol and MoleculeSTM (Smiles/Graph) consistently perform comparably, while other generalist LLM-based methods (e.g., Galactica and Vicuna variants) exhibit significantly weaker performance across all tasks. InstructMol demonstrates strong results as reported in the original paper, though its lack of released weights prevents direct reproducibility. Notably, Token-Mol achieves results that are on par

with MoleculeSTM, indicating that specialized adaptation of MLLMs can substantially narrow the performance gap with task-specific molecular models.

## D.2 PROTEIN PROPERTY PREDICTION

**Experiment setting.** We evaluate models on the TAPE benchmark (Rao et al., 2019) to assess their capability in protein property prediction across six tasks: secondary structure(SS) prediction, contact prediction, homology prediction, fluorescence prediction and stability prediction. Secondary structure and homology prediction are multi-label classification tasks with accuracy used as the evaluation metric. Contact prediction is performed using the precision of the top $L/2$ predicted contacts, where $L$ denotes the sequence length, focusing on medium- and long-range interactions. Fluorescence prediction aims to predict the logarithm of a protein's fluorescence intensity, while stability prediction estimates a proxy for protein stability. Both tasks are evaluated using Spearman's rank correlation coefficient($\rho$).

**Benchmarking Models.** We identify OntoProtein (Zhang et al., 2022a), ProtBERT (Elnaggar et al., 2022), and ProteinDT Liu et al. (2025d). For non-MLLM models, we adopt the results reported in the ProteinDT Liu et al. (2025d).

**Analysis.** As shown in Table D2, traditional baselines such as TAPE Transformer, and MSA Transformer perform moderately, while specialist models like ProtBERT and OntoProtein achieve stronger results. ProteinDT further improve performance across most tasks.

Table D2: Benchmark Results covers six protein property prediction tasks from the TAPE (Rao et al., 2019) benchmark. For non-MLLM models, we adopt the results reported in OntoProtein (Zhang et al., 2022b) and ProteinDT (Liu et al., 2025d).

| Method | Structure | | Evolutionary | | Engineering | |
|---|---|---|---|---|---|---|
| | SS-Q3 ↑ | SS-Q8 ↑ | Contact ↑ | Homology ↑ | Fluorescence ↑ | Stability ↑ |
| LSTM | 0.75 | 0.59 | 0.26 | 0.26 | 0.67 | 0.69 |
| TAPE Transformer | 0.73 | 0.59 | 0.25 | 0.21 | 0.68 | 0.73 |
| ResNet | 0.75 | 0.58 | 0.25 | 0.17 | 0.21 | 0.73 |
| MSA Transformer | - | 0.73 | 0.49 | - | - | - |
| ProtBERT | 0.81 | 0.67 | 0.59 | 0.29 | 0.61 | 0.82 |
| OntoProtein | 0.82 | 0.68 | 0.56 | 0.24 | 0.66 | 0.75 |
| ProteinDT-ProteinCLAP-InfoNCE | 0.8354 | 0.6912 | 0.6011 | 0.3109 | 0.6047 | 0.8110 |
| ProteinDT-ProteinCLAP-EBM-NCE | 0.8310 | 0.6941 | 0.6023 | 0.2865 | 0.6127 | 0.7978 |

Table D3: Protein Property(PP) / Protein Function(PFP) / Proptein-Protein Interaction(PPI) Prediction Tasks.(13 models)

| Model | Pretraining Dataset | Downstream Dataset | Tasks | Metrics | Repo |
|---|---|---|---|---|---|
| ProteinChat | SwissProt | UniProtKB | Specific-category PFP | Acc, MacroF1, WeightedF1 | link |
| ProtLLM | InterPT | PEER | EC prediction | AUPR, $F_{max}$ | link |
| | | PEER | GO term prediction | AUPR, $F_{max}$ | |
| | | Human PPI dataset | PPI prediction | Accuracy | |
| ProLLaMA | UniRef50, InterPro | UniRef50 | PP | Accuracy | link |
| InstructProtein | UniRef100 | DeepLoc | Protein Localization Prediction | Accuracy | link |
| | | GO dataset | PFP | AUPR, Accuracy | |
| | | Hu dataset | PFP | Accuracy | |
| ESM-AA | AlphaFold DB | NetSurfP-2.0, CB513, CASP12, TS115 | Secondary Structure Prediction (SSP) | Accuracy | link |
| | | GO dataset | GO term prediction | AUPR, $F_{max}$ | |
| | | GO dataset | EC prediction | AUPR, $F_{max}$ | |
| BioT5 / BioT5+ | UniRef50 | PEER | (PP) | Accuracy | link |
| | | Yeast, Human | PPI Prediction | Accuracy | |
| ProSST | AlphaFold DB | DeepLoc | Protein Localization Prediction | Accuracy | link |
| | | GO dataset | GO term prediction | $F_{max}$ | |
| | | MIB(TAPE) | PFP | Acc | |
| BioBridge | PrimeKG | SHS27K, SHS148K, STRING | PPI prediction | micro-F1 | link |
| LLaPA | UniProtQA | SHS27K, SHS148K | PPI | micro-F1 | link |
| ProteinDT | SwissProtCLAP | TAPE | PP | varies | link |
| | | GO,DeepLoc | PFP | | |
| OntoProtein | ProteinKG25 | SHS27K, SHS148K, STRING | PPI | micro-F1 | link |
| | | ProteinKG25 | PFP (CAFA) | CAFA score | |
| | | TAPE | PP | varies | |
| SaProt | UniRef50 | FLIP | PFP | Spearman, Acc | link |
| | | DeepLoc | PLP | Accuracy | |
| | | DeepFRI | PAP | $F_{max}$ | |
| | | PEER | PPI | Accuracy | |

Table D4: Protein Generation Tasks.(8 models)

| Model Name | Pretraining Dataset | Downstream Dataset | Generation Tasks | Metrics | Repo |
|---|---|---|---|---|---|
| ProLLaMA | UniRef50,protein2ipr | UniRef50(check) | UPG(sequences) | pLDDT,TM-score,RMSD,SC-Perp | link |
| | | Superfamily | CPG(sequences) | Seq-ident, H-prob,Ref,TM-scores | |
| InstructProtein | UniRef100 | - | CPG(sequences) | pLDDT | link |
| BioT5+ | | N/A | Description-guided Protein Generation | N/A | link |
| DPLM-2 | PDB, SwissProt | N/A | UPG(struct-seq mixed) | sc-TMscore, sc-RMSD,pLDDT,TMscore,F | link |
| | | CAMEO 2022,PDB | Sequence-Conditioned Structure Prediction | RMSD, TMScore | |
| InstructBioMol | Uniref50 | | Description-based Protein Generation | Identity, Alignment, BLOSUM Substitution | link |
| ESM3 | UniRef,MGnify90 | PDB | UPG(seq) | pLDDT, pTM | link |
| FoldGPT link | cAF2DB | CATH4.3 | Backbone inpainting | pLDDT | |
| | | SAbDab | Antibody Design | - | |
| ProteinDT | SwissProtCLAP | SwissProtCLAP | CPG(seq) | retrieval accuracy | link |

Table D5: Protein to Text Generation Tasks.(4 models, UPG:Unconditional Protein Generation, CPG: Conditional Protein Generation)

| Model Name | Pretraining Dataset | Downstream Dataset | Tasks | Metrics | link |
|---|---|---|---|---|---|
| ProtT3 | Swiss-Prot, ProteinKG25 PDB-QA | Swiss-Prot, ProteinKG25 PDB-QA | captioning Protein QA | BLEU, ROUGE, METEOR Exact Match | link |
| BioMedGPT | UniProtQA | UniProtQA | ProteinQA | BLEU, ROUGHE, MEATOR | link |
| BioT5+ | Uniref50 | UniProt | Description Generation | ROUGE-L | link |
| Galactica | | UniProt | Protein Function Description | ROUGE-L | link |

# E  SUMMARY MODEL TABLES

Table E1: Summary of recent representative MLLMs for drug and molecule representation, property prediction, and chemistry-focused tasks.

| Model | Year | Modality | Architecture | Size | Category | Main Task |
|---|---|---|---|---|---|---|
| MolPROP (Rollins et al., 2024) | 2024/05/22 | SMILES, Graph | Encoder-Only | 46M | Property Prediction | Molecular property prediction |
| LLM-MPP (Jin et al., 2025) | 2025/05/20 | SMILES, Graph, Text | Decoder-Only | 8B | Property Prediction | Property prediction interpretability |
| ModuLM (Chen et al., 2025b) | 2025/06/01 | 1D, 2D, 3D, Text | Modular/Encoder | 14B | Property Prediction | Flexible property prediction |
| GIT-Mol (Liu et al., 2024c) | 2023/08/14 | Graph, Image, Text | Encoder-Decoder | 700M | Property Prediction | Property prediction generation |
| PolyLLMem (Zhang & Yang, 2025) | 2025/03/29 | Polymer, Structure, Text | Encoder-Only | 8B | Polymer Informatics | Polymer property prediction |
| Molbind (Xiao et al., 2024c) | 2024/03/13 | Structure, Protein, Text | Encoder-Only | 150M | Property Prediction | Binding affinity prediction |
| BioMedGPT (Luo et al., 2023b) | 2023/08/18 | Protein, Text | Encoder-Decoder | 10B | General-purpose | Biomedical QA multi-modal tasks |
| InstructMol (Cao et al., 2023) | 2023/11/27 | Graph, Text | Encoder-Decoder | 2.2B | General-purpose | Instruction following generation |
| UniMoT (Zhang et al., 2024a) | 2024/08/01 | Graph, Text | Encoder-Decoder | 7B | General-purpose | Generation multi-task |
| Mol-LLM (Lee et al., 2025) | 2025/01/01 | SMILES, Graph, Text | Encoder-Decoder | 7B | General-purpose | Generation multi-task |
| ChemVLM (Li et al., 2025b) | 2024/08/14 | Graph, Image, Text | Encoder-Decoder | 20B | General-purpose | Vision-language tasks |
| Token-Mol (Wang et al., 2025c) | 2024/07/10 | SMILES, 2D/3D | Decoder-Only | N/A | General-purpose | Generative modeling |
| M3LLM (Hu et al., 2024) | 2025/08/03 | Graph, Text | Encoder-Decoder | 1.28B | General-purpose | Generation granularity study |
| ChemCrow (Bran et al., 2023) | 2023/04/11 | Text, Tools | Agent (LLM+Tools) | 100B-1T | Agents & Special Tasks | Chemistry agent |
| ChatMolData (Yu et al., 2024b) | 2024/11/19 | Text, Molecular Data | Agent (LLM+Modules) | 100B-1T | Agents & Special Tasks | Data analysis retrieval |
| ChemToolAgent (Yu et al., 2024a) | 2024/11/11 | Text, Tools | Agent (LLM+Tools) | 100B-1T | Agents & Special Tasks | Tool-use agent |
| ChemAgent (Tang et al., 2025) | 2025/01/11 | Text, Memory | Agent (LLM+Memory) | 100B-1T | Agents & Special Tasks | Agent with memory |
| ChemThinker (Ju et al., 2024) | 2024/09/28 | Text, Tools, Agents | Multi-Agent | 70B | Agents & Special Tasks | Multi-agent reasoning |
| MolPuzzle (Guo et al., 2024) | 2024/01/01 | Multimodal | Special Task | N/A | Puzzle Task | Structure elucidation reasoning |
| MM-RCR (Zhang et al., 2024d) | 2024/07/21 | Text, Graph, SMILES | Encoder-Decoder | 7B | Reaction Condition | Reaction condition recommendation |
| Chem3DLLM (Jiang et al., 2025) | 2025/08/14 | Text, 3D structure | Encoder-Decoder | ∼7B | Drug discovery | Generation |

Table E2: Summary of recent representative MLLMs for protein representation, prediction, and design tasks.

| Model | Date | Modality | Architecture | Size | Category | Main Task |
|---|---|---|---|---|---|---|
| ProteinDT (Liu et al., 2025d) | 2023/02/09 | Sequence, Text | Encoder-Decoder | 220M | Sequence-Text | Protein Design |
| ProtT3 (Liu et al., 2024g) | 2024/05/21 | Sequence, Text | Encoder-Decoder | ∼1.3B | Sequence-Text | QA tasks, Protein captioning |
| ProtCLIP (Zhou et al., 2025a) | 2024/12/28 | Sequence, Text | Encoder-Only | 770M | Sequence-Text | Function prediction |
| OntoProtein (Zhang et al., 2022a) | 2022/01/23 | Sequence, Graph | Encoder-Only | 220M | Sequence-Text | Multi prediction tasks |
| BioMedGPT (Luo et al., 2023a) | 2023/05/26 | Sequence, Text, Graph | Encoder-Decoder | 10B | Sequence-Text | Different QA tasks |
| ProtLLM (Zhuo et al., 2024) | 2024/02/28 | Sequence, Text | Encoder-Decoder | 7B | Sequence-Text | Protein understanding, Generation tasks |
| ProLLaMA (Lv et al., 2025) | 2024/02/26 | Sequence, Text | Encoder-Decoder | 7B | Sequence-Text | Protein understanding, Generation tasks |
| InstructProtein (Wang et al., 2023a) | 2023/10/05 | Sequence, Text, Graph | Decoder-Only | 1.3B / 7B | Sequence-Text | Protein design, Prediction tasks |
| ESM-AA (Zheng et al., 2024) | 2024/03/05 | Sequence, SMILES | Encoder-Only | 35M | Sequence-Text | Classification, Property prediction tasks |
| BioT5 (Pei et al., 2023) | 2023/10/11 | Sequence, SMILES, Text | Encoder-Decoder | 252M | Sequence-Text | Diversity prediction, Generation tasks |
| BioT5+ (Pei et al., 2024) | 2024/02/27 | Sequence, SMILES, Text | Encoder-Decoder | 252M | Sequence-Text | Diversity prediction, Generation tasks |
| Galactica (Taylor et al., 2022) | 2022/11/16 | Sequence, Text | Decoder-Only | 120B | Sequence-Text | Prediction, QA tasks |
| ProteinChat (Huo et al., 2024) | 2024/08/19 | Sequence, Text | Encoder-Decoder | 14B | Sequence-Text | Function prediction, categories |
| ESM3 (Hayes et al., 2025) | 2025/01/16 | Sequence, Text, Structure | Encoder-Decoder | 1.4/7/98B | Geometric-Sequence-Text | Design, Generation tasks |
| proseLM-XL (Ruffolo et al., 2024) | 2024/08/03 | Sequence, Structure | Encoder-Decoder | 6.5B | Geometric-Sequence-Text | Protein Design |
| SaProt (Su et al., 2023a) | 2023/10/01 | Sequence, Structure | Encoder-Only | 650M | Geometric-Sequence-Text | Prediction tasks |
| FoldToken (Gao et al., 2025) | 2024/02/04 | Sequence, Structure | Encoder-Decoder | 280M | Geometric-Sequence-Text | Reconstruction, Antibody Design |
| Evolla (Zhou et al., 2025d) | 2025/01/05 | Sequence, Text, Structure | Encoder-Decoder | 80B | Geometric-Sequence-Text | Diverse QA tasks |
| DPLM-2 (Wang et al., 2024b) | 2024/10/17 | Sequence, Structure | Encoder-Decoder | 150/650M | Geometric-Sequence-Text | Protein generation, Folding |
| ProTokens (Lin et al., 2023a) | 2023/11/27 | Sequence, Structure | Encoder-Decoder | 7B | Geometric-Sequence-Text | Protein Design |
| ProSST (Li et al., 2024a) | 2024/04/15 | Sequence, Structure | Encoder-Decoder | 110M | Geometric-Sequence-Text | Prediction tasks |
| ProteinGPT (Xiao et al., 2024d) | 2024/08/21 | Sequence, Text, Structure | Encoder-Decoder | 10B | Geometric-Sequence-Text | Protein QA Protein understanding |
| ProtChatGPT (Wang et al., 2025a) | 2024/02/15 | Sequence, Text, Structure | Encoder-Decoder | 13B | Geometric-Sequence-Text | Protein QA, Protein understanding |
| STELLA (Xiao et al., 2025b) | 2025/06/04 | Sequence, Text, Structure | Encoder-Decoder | ∼9B | Geometric-Sequence-Text | Structure understanding, QA tasks |
| InstructBioMol (Zhuang et al., 2024) | 2024/10/10 | Sequence, Text, SMILES, Structure | Encoder-Decoder | ∼7B | Geometric-Sequence-Text | Protein Design, QA tasks |
| BioBRIDGE (Wang et al., 2023b) | 2023/10/05 | Sequence, Graph, Text | Encoder-Only | ∼3B | Special-case | PPI Prediction |
| LLaPA (Zhou et al., 2025c) | 2024/09/26 | Sequence, Graph, Text | Encoder-Decoder | ∼10B | Special-case | PPI Prediction |
| MolBind (Xiao et al., 2024b) | 2024/03/13 | Text, SMILES, Graph, Structure | Encoder-Only | N/A | Special-case | Retrieval tasks |
| BioTranslator (Xu et al., 2023) | 2023/02/10 | Text, Gene, Sequence, Graph | Encoder-Only | 230M | Special-case | Modal Translator |

Table E3: Representative MLLMs for gene function prediction, regulatory genomics, and multimodal biological tasks.

| Model | Date | Modality | Architecture | Size | Category | Main Task |
|---|---|---|---|---|---|---|
| GeneChat (Dhanasekar et al., 2025) | 2025/06/05 | DNA, Text | DNABERT-2 + Adaptor + Vicuna-13B | ~13B | Function Prediction | Free-text gene function generation |
| ChatNT (Richard et al., 2024) | 2024/04/30 | DNA, RNA, Protein, Text | Nucleotide Transformer + Perceiver + Vicuna-7B | ~7B | Multi-task Genomics | Multimodal sequence Language Q&A |
| LLaMA-Gene (Liang, 2024a) | 2024/11/30 | DNA, Protein, Text | LLaMA3-7B | ~7B | Multi-task Genomics | Gene classification Structure prediction MSA Function prediction Regression |
| OmniCellTOSG (Zhang et al., 2025a) | 2025/04/02 | RNA, Text | DeBERTa+DNAGPT+ ProtGPT2+GAT | ~16B | Multi-task Genomics | Predict cellular states Predict cell types |
| Geneverse (Liu et al., 2024e) | 2024/07/21 | DNA, Protein, Text, Figure | Multi-model LLM/MLLM collection | ~7/8/13B | Multi-task Genomics | Multi-modal gene/protein tasks |
| GenoMAS (Liu et al., 2025a) | 2025/07/08 | DNA, RNA, Text | LLM Agents | N/A | Gene Expression Analysis | (Un)conditional GTA Report Generation |
| cGSA (Wang et al., 2025e) | 2025/06/04 | DNA, Text | LLaMA 3.1-70B | ~70B | Gene Expression Analysis | Gene pathway finding |
| GTA (Honig et al., 2024) | 2024/10/02 | DNA, Text | Sei Encoder + Token Alignment + Llama3-8B | ~8B | Gene Expression Analysis | Long-range gene expression modeling |
| LLM4GRN (Afonja et al., 2024) | 2024/10/21 | RNA, Text | LLaMA3.1-70B | ~70B | Regulatory Genomics | Gene regulatory network discovery |
| GeneBERT (Mo et al., 2021) | 2021/10/11 | DNA (1D), TF-Region (2D) | BERT+ Swin Transformer | ~110M | Regulatory Genomics | Multi-modal self-supervised pre-training |
| GeneCompass (Yang et al., 2024b) | 2023/09/28 | RNA, Text | Transformer | N/A | Regulatory Genomics | GRN inference |

Table E4: Summary of recent representative LLMs and MLLMs for material discovery, property prediction, and design tasks.

| Model | Date | Modality | Architecture | Size | Category | Main Task |
|---|---|---|---|---|---|---|
| CrystaLLM (Antunes et al., 2024) | 2023/07/10 | Text | Decoder-Only | 25/200M | Crystal Structure | Generate crystal structures |
| LLMatDesign (Jia et al., 2024) | 2024/06/19 | Text | LLM Agent | N/A | Autonomous Discovery | Autonomous materials discovery |
| FlowLLM (Sriram et al., 2024) | 2024/10/30 | Text | LLM+RFM | N/A | Material Design | Generate stable novel materials |
| GenMS (Yang et al., 2024a) | 2024/09/10 | Text, Graph | LLM+Diffusion | N/A | Crystal Generation | Low-energy crystal structure generation |
| Mat2Seq (Yan et al., 2024) | 2024/12/01 | Text, Graph | Encoder-Decoder | 25/200M | Property Prediction | Crystal sequence representation |
| CrystaltextLLM (Gruver et al., 2024) | 2024/02/06 | Text | Encoder-Decoder | ~70B | Stability Prediction | Generate stable materials |
| ChatGPTMaterial (Deb et al., 2024) | 2024/02/12 | Text | Decoder-Only | 11B | Material Design | Suggest material compositions |
| ICGPT (Liu et al., 2025b) | 2024/04/22 | Text | Transformer | N/A | Property Prediction | Accurate material property prediction |
| ELLM (Grandi et al., 2025) | 2024/04/23 | Text | Encoder-Decoder | N/A | Material Selection | Expert recommendations for materials |
| ElaTBot (Liu et al., 2024d) | 2024/11/19 | Text, Quantitative Data | Llama2-7B | ~7B | Material Discovery | (Details TBD) |
| CrossMatAgent (Tian et al., 2025) | 2025/03/25 | Text,Image | Agent | N/A | Material Discovery | Multi-agent material design framework |
| AutoMEX (Fan et al., 2025) | 2025/03/– | Text,3D Document Structure Data | Agent | N/A | Material Selection | Autonomous material extrusion workflow |
| LLM-Fusion (Boyar et al., 2025) | 2024/12/19 | Text, SMILES, Fingerprints | Encoder-Decoder | N/A | Property Prediction | Multimodal property prediction |
| Cephalo (Buehler, 2024) | 2024/05/29 | Image, Text | VLM | ~600M | Bio-Inspired Design | Analyze bio-inspired materials |
| MaCBench (Alampara et al., 2024) | 2024/10/08 | Text, Image | VLM | N/A | Material Discovery | Evaluate multimodal models' performance |
| FMMD (Pyzer-Knapp et al., 2025) | 2024 | Text, Image | Fusion Model | N/A | Material Prediction | Scalable property prediction |
| MatterGPT (Chen et al., 2024b) | 2024/08/14 | Text | Transformer | 80M | Property Prediction | Generate solid-state materials |

Table E5: Representative MLLMs for biomedical science.

| Model | Date | Modality | Architecture | Size | Main Tasks |
|---|---|---|---|---|---|
| GenoMAS (Liu et al., 2025a) | 2025/07/08 | DNA, RNA, Text | LLM agents | N/A | Gene expression analysis |
| cGSA (Wang et al., 2025e) | 2025/06/04 | DNA, Text | LlaMA 3.1-70B | ~70B | Gene pathway findiing |
| LLM4GRN (Afonja et al., 2024) | 2024/10/21 | RNA, Text | LLaMA3.1-70B | ~70B | Gene regulatory networks discovery |
| GeneCompass (Yang et al., 2024b) | 2023/09/28 | RNA, Text | Transformer | N/A | Gene Regulatory Network inference |
| Geneverse (Liu et al., 2024e) | 2024/07/21 | DNA, Protein Text, Figure | Multi-model LLM/MLLM collection | ~7/8/13B | Multi-modal gene/protein tasks |
| BioMedGPT (Luo et al., 2023a) | 2024/11/25 | Natural Language Molecular Graphs Protein Sequences | BioMedGPT-LM+ Multimodal encoder | ~10B | Protein Question Answering Molecule Question Answering |
| LLaMA-Gene (Liang, 2024a) | 2024/11/30 | DNA, Protein, Text | LLaMA3-7B | ~7B | Gene classification Gene structure prediction Multiple sequence analysis Function prediction |
| OmniCellTOSG (Zhang et al., 2025a) | 2025/04/02 | RNA, Text | DeBERTa+DNAGPT +ProtGPT2+GAT | ~16B | Cellular States Prediction Cell Type Prediction |
| mSTAR (Xu et al., 2025) | 2024/07/22 | pathological images, RNA-seq, Text | CLIP | Varies | Survival prediction Diagnosis Molecule prediction Report generation |
| ST-ALign (Lin et al., 2024) | 2024/11/25 | pathological images, gene | Image encoder + Gene encoder | N/A | Spatial clustering identification Spot Gene Expression Prediction |
| spEMO (Liu et al., 2025e) | 2025/01/13 | Pathological images spatial multi omics | PFM+LLM | N/A | Spatial domain identification Disease Prediction Report Generation |
| SpaLLM (Li et al., 2025c) | 2025/07/03 | Single-cell transcriptome data, Multi-omics data | LLM+omics encoder+GNN | N/A | Region Identification |

# F Summary Dataset Tables of MLLMs for Science

Table F1: Summary of pretraining / instruction-tuning datasets for MLLMs in molecular tasks.

| Datasets | Year | Modality | Tasks | Source | Application | Stage |
|---|---|---|---|---|---|---|
| PubChem (77M SMILES) | – | SMILES, Text | MLM, MTR, caption/retrieval | Source | Rollins et al. (2024) Liu et al. (2024c) Le et al. (2024) Cao et al. (2023) Zhang et al. (2024a) Livne et al. (2024) Chen et al. (2025b) Jin et al. (2025) | Pretraining |
| ChEBI-20 | 2021 | SMILES, Text | Captioning, generation | Source | Liu et al. (2024c) Zhang et al. (2024a) Lee et al. (2025) Cao et al. (2023) | Pretraining |
| ZINC | – | SMILES | Language modeling, generation | Source | Livne et al. (2024) | Pretraining |
| USPTO (full/50k) | 2012/2017 | Reaction SMILES, Text | FS/RS/RP reaction modeling | Source (full) Source (full) Source (50k) | Lee et al. (2025) Zhang et al. (2024a) | Pretraining/Instr. |
| Mol-Instructions | 2023 | Text, SMILES, Graph | FS, RS, RP, caption-guided gen | Source | Lee et al. (2025) Zhang et al. (2024a) | Instruction |
| SMolInstruct | 2024 | Text, SMILES, Graph | FS, RS, RP, generation | Source | Lee et al. (2025) | Instruction |
| PCdes | – | Molecule, Text | Retrieval (M2T/T2M) | Source | Zhang et al. (2024a) | Instruction |
| MoMu | 2022 | Molecule, Text | Cross-modal retrieval | Source | Zhang et al. (2024a) | Instruction |
| Molecule3D | 2021 | 3D | Conformations Graph–3D alignment | Source Source | Xiao et al. (2024c) | Pretraining |
| GEOM | 2020 | 3D | Conformations Graph–3D alignment | Source | Xiao et al. (2024c) | Pretraining |
| PDBBind | 2016 | Protein pockets, 3D | Conf.–Protein alignment | Source | Xiao et al. (2024c) | Pretraining |
| CrossDock | 2019 | Protein pockets, 3D | Conf.–Protein alignment | Source | Xiao et al. (2024c) | Pretraining |
| DrugBank | – | SMILES, Text (properties) | Molecular relational learning | Source | Chen et al. (2025b) | Pretraining |
| L+M-24 | 2024 | Image, Text | Captioning (Mol2Lang) | Source | Tran et al. (2024) | Pretraining |
| Chem Exam | 2024–2025 | Image, Text | OCR, VQA, Chem QA | Source | Li et al. (2025b) | Pretraining |
| Chem OCR | 2024–2025 | Image, Text | OCR, VQA, Chem QA | Source | Li et al. (2025b) | Pretraining |
| Web-Chem | 2024–2025 | Image, Text | OCR, VQA, Chem QA | Source | Li et al. (2025b) | Pretraining |
| PubMed abstracts | – | Text (biomedical) | Domain LM pretraining | Source | Luo et al. (2022) | Pretraining |

Table F2: Summary of downstream task datasets for MLLMs in molecular tasks.

| Datasets | Year | Modality | Tasks | Source | Application | Stage |
|---|---|---|---|---|---|---|
| ESOL (LogS) | 2012 | SMILES, Graph | Regression (solubility) | source | Rollins et al. (2024) Jin et al. (2025) Lee et al. (2025) Le et al. (2024) | Downstream |
| FreeSolv | 2014 | SMILES, Graph | Regression (hydration free energy) | source | Rollins et al. (2024) Jin et al. (2025) Chen et al. (2025b) | Downstream |
| Lipophilicity (Lipo) | 2016 | SMILES, Graph | Regression (logD/logP) | source | Rollins et al. (2024) Jin et al. (2025) Lee et al. (2025) | Downstream |
| QM7 | 2011 | SMILES, Graph | Regression (atomization energy) | source | Rollins et al. (2024) Jin et al. (2025) | Downstream |
| QM9 | 2014 | SMILES, Graph | Regression (HOMO/LUMO etc.) | source | Cao et al. (2023) Lee et al. (2025) | Downstream |
| BBBP | 2018 | SMILES, Graph | Classification (BBB) | source | Rollins et al. (2024) Jin et al. (2025) Lee et al. (2025) Le et al. (2024) | Downstream |
| BACE | 2016 | SMILES, Graph | Classification (binding) | source | Rollins et al. (2024) Jin et al. (2025) Lee et al. (2025) Le et al. (2024) | Downstream |
| ClinTox | 2018 | SMILES, Graph | Classification (toxicity) | source | Rollins et al. (2024) Jin et al. (2025) Lee et al. (2025) Le et al. (2024) | Downstream |
| Tox21 | 2014 | SMILES, Graph | Multi-task toxicity | source | Liu et al. (2024c) Zhang et al. (2024a) Le et al. (2024) | Downstream |
| ToxCast | 2013 | SMILES, Graph | Multi-task toxicity | source | Liu et al. (2024c) Zhang et al. (2024a) | Downstream |
| HIV | 2014 | SMILES, Graph | Classification (anti-HIV) | source | Lee et al. (2025) Le et al. (2024) | Downstream |
| SIDER | 2015 | SMILES, Graph | Multi-label side effects | source | Liu et al. (2024c) Lee et al. (2025) Le et al. (2024) | Downstream |
| MUV | 2013 | SMILES, Graph | Virtual screening | source | Le et al. (2024) | Downstream |
| ChEBI-20 | 2021 | SMILES, Text | Captioning, generation | source | Liu et al. (2024c) Lee et al. (2025) Zhang et al. (2024a) Le et al. (2024) | Downstream |
| L+M-24 | 2024 | Image, Text | Captioning | source | Tran et al. (2024) | Downstream |
| PubChem Captions | – | Image, SMILES, Text | Captioning, Image→SMILES | source | Liu et al. (2024c) | Downstream |
| USPTO-50k | 2017 | Reaction SMILES, Text | FS, RS, RP | source | Lee et al. (2025) Cao et al. (2023) | Downstream |
| RetroBench | 2024 | Reaction network | Multi-step retrosynthesis | source | Kang et al. (2025) | Downstream |
| ORDERly | 2024 | Reactions | OOD reaction evaluation | source | Lee et al. (2025) | Downstream |
| AqSolDB | 2019 | SMILES | OOD solubility evaluation | source | Lee et al. (2025) | Downstream |
| ChEMBL-02 | 2020 | Pairwise molecules | Molecule optimization | source | Le et al. (2024) | Downstream |
| PCdes | – | Molecule, Text | Retrieval (M2T/T2M) | source | Zhang et al. (2024a) | Downstream |
| MoMu | 2022 | Molecule, Text | Cross-modal retrieval | source | Zhang et al. (2024a) | Downstream |
| ZhangDDI | 2017 | SMILES, Graph | Drug–drug interaction | source | Chen et al. (2025b) | Downstream |
| ChChMiner | 2018 | SMILES, Graph | Drug–drug interaction | source | Chen et al. (2025b) | Downstream |
| DeepDDI | 2018 | SMILES, Graph | Drug–drug interaction | source | Chen et al. (2025b) | Downstream |
| TWOSIDES | 2012 | SMILES, Graph | Drug–drug interaction | source | Chen et al. (2025b) | Downstream |
| MNSol | 2020 | SMILES, Graph | Solute–solvent interaction | source | Chen et al. (2025b) | Downstream |
| CompSol | 2017 | SMILES, Graph | Solute–solvent interaction | source | Chen et al. (2025b) | Downstream |
| Abraham | 2010 | SMILES, Graph | Solute–solvent interaction | source | Chen et al. (2025b) | Downstream |
| CombiSolv | 2021 | SMILES, Graph | Solute–solvent interaction | source | Chen et al. (2025b) | Downstream |
| CombiSolv-QM | 2021 | SMILES, Graph (QM) | Solute–solvent interaction | source | Chen et al. (2025b) | Downstream |
| Chromophore | 2020 | SMILES, Graph | Chromophore–solvent interaction | source | Chen et al. (2025b) | Downstream |

Table F3: Summary of pretraining / instruction-tuning datasets for MLLMs in protein tasks.

| Datasets | Year | Modality | Tasks | Source | Application | Stage |
|---|---|---|---|---|---|---|
| SwissProt | 2000 | Sequence, Text | Sequence–text alignment, Captioning | Source | Liu et al. (2025c) Liu et al. (2024g) Zhou et al. (2025a) Huo et al. (2024) Zhou et al. (2025d) | Pretraining |
| TrEMBL | 2000 | Sequence, Text | Sequence–text alignment | Source | Zhou et al. (2025a) Zhou et al. (2025d) | Pretraining |
| ProtAnno-S | 2024 | Sequence, Text | Contrastive alignment (sparse, curated) | Source | Zhou et al. (2025a) | Pretraining |
| ProtAnno-D | 2024 | Sequence, Text | Contrastive alignment (dense, auto) | Source | Zhou et al. (2025a) | Pretraining |
| ProteinKG25 | 2022 | Sequence, Graph, Text | KG-enhanced pretraining | Source | Zhang et al. (2022c) Liu et al. (2024g) | Pretraining |
| PrimeKG | 2023 | Graph, Text | Biomedical KG bridging | Source | Wang et al. (2023b) | Pretraining |
| UniRef50 | 2007 | Sequence | Language modeling corpus | Source | Lv et al. (2025) | Pretraining |
| UniRef90 | 2007 | Sequence | Language modeling corpus | Source | Wang et al. (2024b) | Pretraining |
| AlphaFold DB | 2022 | Structure (3D) | Structure-aware pretraining | Source | Su et al. (2023a) Zheng et al. (2024) Hayes et al. (2025) | Pretraining |
| PDB | 2000 | Structure (3D) | Structure and token pretraining | Source | Wang et al. (2024b) Lin et al. (2023a) | Pretraining |
| PDBbind (v2019) | 2019 | Structure, Binding | Binding-aware pretraining | Source | Zheng et al. (2024) | Pretraining |
| S2ORC | 2020 | Text (scholarly) | Biomedical text pretraining | Source | Luo et al. (2023a) | Pretraining |
| PubMed abstracts | 1996 | Text (biomedical) | Biomedical text pretraining | Source | Luo et al. (2023a) Zhuo et al. (2024) Pei et al. (2024) | Pretraining |
| bioRxiv | 2013 | Text (preprints) | Biomedical text pretraining | Source | Pei et al. (2024) | Pretraining |
| PubChem | 2004 | SMILES, Text | Chem–structure pretraining | Source | Pei et al. (2023) Pei et al. (2024) | Pretraining |
| ChEMBL | 2012 | SMILES, Bioactivity | Chem–structure pretraining | Source | Zheng et al. (2024) Pei et al. (2023) | Pretraining |
| ZINC (ZINC15) | 2015 | SMILES | Generative pretraining | Source | Pei et al. (2023) Pei et al. (2024) | Pretraining |
| InterPT (instruction set) | 2024 | Sequence, Text | Protein–text instruction pretraining | Source | Zhuo et al. (2024) | Instruction |
| ProteinChat Corpus | 2024 | Sequence, Text | Instruction/QA pretraining | Source | Huo et al. (2024) | Instruction |
| SwissProtCLAP | 2023 | Sequence, Text | Sequence–text alignment | Source | Liu et al. (2025c) | Pretraining |

Table F4: Summary of downstream task datasets for MLLMs in protein tasks.

| Datasets | Year | Modality | Tasks | Source | Application | Stage |
|---|---|---|---|---|---|---|
| TAPE | 2019 | Sequence, Structure | SS, Contact, Homology, Fluorescence, Stability | Source | Liu et al. (2025c) Zhang et al. (2022c) Zhuo et al. (2024) Zheng et al. (2024) Wang et al. (2023a) Ruffolo et al. (2024) Su et al. (2023a) | Downstream |
| DeepLoc | 2017 | Sequence, Text | Subcellular localization | Source | Zhou et al. (2025a) Wang et al. (2023a) | Downstream |
| Solubility (DeepSol) | 2017 | Sequence | Solubility prediction | Source | Pei et al. (2023) | Downstream |
| Localization | 2017 | Sequence | Membrane/soluble classification | Source | Pei et al. (2023) | Downstream |
| SwissProt | 2000 | Sequence, Text | Function description classification | Source | Wang et al. (2023a) Huo et al. (2024) | Downstream |
| CASP15 | 2022 | Structure | Protein folding | Source | Hayes et al. (2025) | Downstream |
| CB513 | 1999 | Sequence | Secondary structure prediction | Source | Su et al. (2023a) Li et al. (2024a) | Downstream |
| SCOPe | 2014 | Structure | Fold/superfamily classification | Source | Lv et al. (2025) Ruffolo et al. (2024) Li et al. (2024a) | Downstream |
| TAPE Stability | 2019 | Sequence | Stability prediction | Source | Ruffolo et al. (2024) | Downstream |
| TAPE Contact | 2019 | Structure | Contact map prediction | Source | Su et al. (2023a) Wang et al. (2023a) | Downstream |
| STRING | 2021 | Graph (PPI) | PPI classification | Source | Zhang et al. (2022c) Zhuo et al. (2024) Wang et al. (2023a) Wang et al. (2023b) Zhou et al. (2025c) | Downstream |
| SHS27k | 2019 | Sequence, Graph | PPI classification | Source | Zhang et al. (2022c) Zhuo et al. (2024) Wang et al. (2023a) Wang et al. (2023b) | Downstream |
| SHS148k | 2019 | Sequence, Graph | PPI classification | Source | Zhang et al. (2022c) Zhuo et al. (2024) Wang et al. (2023a) Wang et al. (2023b) | Downstream |
| BioGRID | 2003 | Graph | PPI classification | Source | Zhou et al. (2025c) | Downstream |
| PPI (Yeast, Human) | 2019 | Sequence, Graph | PPI classification | Source | Pei et al. (2023) | Downstream |
| BioSNAP | 2018 | Sequence, Graph | DTI, PPI prediction | Source | Pei et al. (2023) | Downstream |
| DMS ($\beta$-lac, AAV, Thermo, Flu, Sta) | 2018 | Sequence | Mutational effect prediction | Source | Zhou et al. (2025a) | Downstream |
| ProteinGym | 2023 | Sequence | Mutational effect prediction | Source | Hayes et al. (2025) Su et al. (2023a) Li et al. (2024a) | Downstream |
| PubMedQA | 2019 | Text | Biomedical QA | Source | Luo et al. (2023a) Taylor et al. (2022) Xu et al. (2023) | Downstream |
| MedMCQA | 2022 | Text | Biomedical QA | Source | Luo et al. (2023a) Taylor et al. (2022) | Downstream |
| USMLE | 2020 | Text | Medical exam QA | Source | Luo et al. (2023a) Taylor et al. (2022) | Downstream |
| UniProtQA | 2023 | Sequence, Text | Protein QA | Source | Luo et al. (2023a) Taylor et al. (2022) Xu et al. (2023) | Downstream |
| ProteinQA benchmark | 2024 | Sequence, Text | Protein QA | Source | Huo et al. (2024) Xiao et al. (2024d) Wang et al. (2025a) Xiao et al. (2025b) | Downstream |
| PDB-QA | 2024 | Structure, Text | Protein QA | Source | Liu et al. (2024g) | Downstream |
| MMLU-bio | 2021 | Text | Multitask biomedical QA | Source | Taylor et al. (2022) | Downstream |
| ChEBI-20 | 2019 | Molecule, Text | Molecule QA, Captioning | Source | Luo et al. (2023a) Pei et al. (2023) | Downstream |
| ChemProt | 2019 | Text | Relation extraction | Source | Pei et al. (2023) | Downstream |
| BindingDB | 2007 | Sequence, SMILES | Binding prediction | Source | Zheng et al. (2024) Pei et al. (2023) Xiao et al. (2024b) | Downstream |
| MoleculeNet | 2018 | Molecule | Property prediction | Source | Zheng et al. (2024) Taylor et al. (2022) | Downstream |
| USPTO | 2019 | SMILES, Text | Reaction prediction | Source | Taylor et al. (2022) | Downstream |
| PubChem BioAssay | 2014 | SMILES, Text | Retrieval | Source | Xiao et al. (2024b) | Downstream |
| SAbDab | 2014 | Structure | Antibody design | Source | Gao et al. (2025) | Downstream |
| Inverse folding sets | 2019 | Sequence, Structure | Inverse folding | Source | Lin et al. (2023a) | Downstream |
| Protein design benchmarks | 2024 | Sequence, Structure | Protein generation, Design | Source | Hayes et al. (2025) Zhou et al. (2025d) Zhuang et al. (2024) | Downstream |

Table F5: Summary of pretraining / instruction-tuning datasets for MLLMs in gene tasks.

| Datasets | Year | Modality | Tasks | Source | Application | Stage |
|---|---|---|---|---|---|---|
| NCBI Gene | 2005 | DNA, Text | Function modeling | source | Dhanasekar et al. (2025) | Pretraining |
| NT | 2023 | DNA | Sequence classification | source | Richard et al. (2024) | Pretraining |
| BEND | 2022 | DNA | Regulatory element classification | source | Richard et al. (2024) | Pretraining |
| AgroNT | 2023 | DNA | Plant genomics tasks | source | Richard et al. (2024) | Pretraining |
| ChromTransfer | 2022 | DNA | Regulatory element transfer | source | Richard et al. (2024) | Pretraining |
| ATAC-seq fetal atlas | 2020 | DNA, TF-region | Chromatin accessibility | source | Mo et al. (2021) | Pretraining |
| Sei | 2022 | DNA, Chromatin | Epigenomic feature extraction | source | Honig et al. (2024) | Pretraining |
| SwissProt | 1986 | Protein | Protein sequence modeling | source | Liang (2024a) | Pretraining |
| TrEMBL | 1996 | Protein | Protein sequence modeling | source | Liang (2024a) | Pretraining |
| S2ORC | 2020 | Text | Scientific text modeling | source | Liang (2024a) | Pretraining |
| scCompass-126M | 2024 | RNA | Cross-species modeling | source | Yang et al. (2024b) | Pretraining |
| Ensembl GRCh38 | 2013 | DNA | Genomic sequences | source | Liu et al. (2024e) | Pretraining |
| GTEx v8 | 2015 | RNA | Expression profiles | source | Liu et al. (2024e) | Pretraining |
| UniProt | 2023 | Protein | Protein sequences | source | Liu et al. (2024e) | Pretraining |
| PubMed abstracts | 1996 | Text | Biomedical language modeling | source | Liu et al. (2024e) | Pretraining |

Table F6: Summary of downstream task datasets for MLLMs in gene tasks.

| Datasets | Year | Modality | Tasks | Source | Application | Stage |
|---|---|---|---|---|---|---|
| NCBI Gene | 2005 | DNA, Text | Function prediction | source | Dhanasekar et al. (2025) | Downstream |
| NT | 2023 | DNA | Sequence classification | source | Richard et al. (2024) | Downstream |
| BEND | 2022 | DNA | Regulatory element classification | source | Richard et al. (2024) | Downstream |
| AgroNT | 2023 | DNA | Plant genomics tasks | source | Richard et al. (2024) | Downstream |
| ChromTransfer | 2022 | DNA | Regulatory element transfer | source | Richard et al. (2024) | Downstream |
| DeepSTARR | 2019 | DNA | Enhancer activity prediction | source | Richard et al. (2024) | Downstream |
| APARENT2 | 2022 | RNA | Polyadenylation prediction | source | Richard et al. (2024) | Downstream |
| Saluki | 2022 | RNA | RNA degradation prediction | source | Richard et al. (2024) | Downstream |
| GM12878 | 2012 | RNA | Expression prediction | source | Honig et al. (2024) | Downstream |
| Geuvadis | 2013 | RNA | Expression prediction | source | Honig et al. (2024) | Downstream |
| GenoTEX | 2025 | DNA, RNA | Gene–trait association | source | Liu et al. (2025a) | Downstream |
| GEO | 2002 | RNA | Expression prediction | source | Liu et al. (2025a) | Downstream |
| TCGA | 2008 | RNA, DNA | Expression prediction | source | Liu et al. (2025a) | Downstream |
| Curated gene sets (102) | 2025 | Gene sets | Pathway enrichment | source | Wang et al. (2025e) | Downstream |
| Case studies (melanoma, breast cancer) | 2025 | RNA, Text | Disease-specific analysis | source | Wang et al. (2025e) | Downstream |
| UniProt | 2023 | Protein | Function prediction | source | Liang (2024a) | Downstream |
| Pfam | 1997 | Protein | Domain classification | source | Liang (2024a) | Downstream |
| InterPro | 2000 | Protein | Domain classification | source | Liang (2024a) | Downstream |
| PBMC-ALL | 2017 | RNA | GRN inference | source | Afonja et al. (2024) | Downstream |
| PBMC-CTL | 2017 | RNA | GRN inference | source | Afonja et al. (2024) | Downstream |
| BoneMarrow | 2019 | RNA | GRN inference | source | Afonja et al. (2024) | Downstream |
| OmniCellTOSG | 2025 | scRNA-seq, Text | Cellular state prediction | source | Zhang et al. (2025a) | Downstream |
| HCA | 2017 | scRNA-seq | Cross-species GRN inference | source | Yang et al. (2024b) | Downstream |
| MCA | 2018 | scRNA-seq | Cross-species GRN inference | source | Yang et al. (2024b) | Downstream |
| Tabula Sapiens | 2022 | scRNA-seq | Cross-species GRN inference | source | Yang et al. (2024b) | Downstream |
| GO annotation | 2000 | DNA, Text | Function prediction | source | Liu et al. (2024e) | Downstream |
| UniProt | 2002 | Protein | Protein classification | source | Liu et al. (2024e) | Downstream |
| GTEx v8 | 2015 | RNA | Expression prediction | source | Liu et al. (2024e) | Downstream |

