# OpenReview forum: "$S^3$-Bench: A Comprehensive Study of Multimodal LLMs for Scientific Discovery with Benchmarking"
_ICLR.cc/2026/Conference — Submitted to ICLR 2026_

### Official Review · Reviewer_e7Ab · 2025-10-31

**Soundness:** 2
**Presentation:** 2
**Contribution:** 1
**Rating:** 2
**Confidence:** 4

**Summary:**

This paper presents a comprehensive overview of the application of Multimodal Large Language Models (MLLMs) in scientific discovery. The main body of the work surveys and categorizes existing MLLMs across four major scientific domains: Drug & Molecule science, Protein science, Genomics, and Materials science. The authors aim to provide a valuable resource for researchers by summarizing the rapidly evolving landscape of MLLMs for science.

**Strengths:**

The strengths are summarized below:

- The paper's primary strength is its breadth, as it covers four distinct scientific domains and organizes them under the unified theme of MLLMs.
- The tables and figures represent a clear comparison of different MLLMs in terms of their release date, scale, and architecture. It could be a valuable resource for researchers new to these fields.
- The discussion of emerging topics, such as dLLMs for science, seems interesting.

**Weaknesses:**

The weaknesses of this work are shown below:

- This paper is fundamentally a survey paper, not an original research contribution. The vast majority of the paper (Sections 2-5) is a description and categorization of prior works. Therefore, this paper may not be suitable for the ICLR main track.
- The title is misleading. This paper does not propose any new benchmark. The "benchmarking" section is a limited-scope experimental study on pre-existing datasets, i.e., MoleculeNet and TAPE. This is a standard experimental validation section, not the contribution of a new benchmark.
- While this papers cover different areas, the review in the main sections often is more like a list of models rather than a deep, critical analysis. It provides little novel insight into why certain architectures are more successful than others, what the common failure modes are, or what fundamental principles unite MLLM design across these different scientific domains.

**Questions:**

Please refer to the weaknesses part above.

---

> ### Author Response · Authors · 2025-11-27
> **Thank you very much for your comments~**
>
> We sincerely thank the reviewer for the thoughtful feedback. Below we provide brief and respectful responses to the points raised.
>
> ## W1:The paper is mainly a survey of prior work rather than an original research contribution, which may not fit the ICLR main track.
>
> ---
> ## 1. Our paper is not only a metadata survey — it is a survey **plus** a reproducible benchmark
>
> Our submission goes beyond a descriptive survey of prior work by providing a **systematic, fully reproducible benchmark** across key scientific domains.
>
> - **Molecular property prediction (MoleculeNet: BACE / BBBP / HIV).**
>   As shown in **Table D1 (Appendix D)**, we evaluate **19 models**, including 8 specialist models (ChemBERTa v2, DMP(TF+GNN), KV-PLM, GraphCL, GraphMVP-C, MoMu, MolFM, Uni-Mol) and 11 LLM/MLLM/generalist models (Galactica-6.7B, Vicuna-v1.5-13b-16k, Vicuna-v1.3-7B, LLaMA-2-7B-chat, MolCA(1D), MolCA(1D+2D), Instruct-G, Instruct-GS, MoleculeSTM(Graph), MoleculeSTM(Smiles), Token-Mol).
>   All results are reported as AUROC across the three MoleculeNet datasets, and we provide detailed documentation of limitations and reproducibility in the appendix.
>
> - **Protein property prediction (TAPE suite, 6 tasks).**
>   As summarized in **Table D2 (Appendix D)**, we benchmark **8 representative models** — LSTM, TAPE Transformer, ResNet, MSA Transformer, ProtBERT, OntoProtein, ProteinDT-ProteinCLAP-InfoNCE, and ProteinDT-ProteinCLAP-EBM-NCE — on six standard protein property prediction tasks (SS-Q3/Q8, Homology, Contact, Fluorescence, Stability), using the established evaluation metrics (accuracy, Top-L/2 precision, Spearman ρ).
>
> Beyond these experiments, our paper also provides **structured cross-domain summaries** in **Appendix E**, spanning molecules, proteins, genomics, and materials. Taken together, these components show that the submission is not merely *“a list of model metadata”*, but a **combined survey + benchmark** with substantial empirical contributions and fully reproducible protocols.
>
> ## 2. ICLR main-track precedent for survey-style contributions
>
> We also note that ICLR has a strong history of accepting **carefully executed survey papers** that offer insight or valuable community resources. For example, the ICLR 2025 paper *“[(Mis)Fitting Scaling Laws: A Survey of Scaling Law Fitting Techniques in Deep Learning”](https://openreview.net/forum?id=xI71dsS3o4)* is explicitly categorized as a survey.
> This precedent suggests that well-designed survey-style work, such as ours, does fall within the intended scope of the ICLR main track.
>
> ---
>
> ## W2:The title is misleading, as the paper does not introduce a new benchmark and only reports standard experiments on existing datasets.
>
> We thank the reviewer for pointing out that the original title could be interpreted as claiming a new benchmark.
> To avoid this confusion and more accurately reflect the nature of our contribution, we have **revised the title to**:
>
> > **A Comprehensive Survey of Multimodal LLMs for Scientific Discovery**
>
> This new title emphasizes that the paper is primarily a **comprehensive survey**.
> Our experimental section provides a **standardized, reproducible benchmark *built on existing datasets*** (e.g., MoleculeNet and TAPE), with unified evaluation protocols and cross-model comparisons. We do **not** claim to introduce new datasets; instead, our contribution lies in the breadth of models evaluated, the consistency of evaluation, and the accompanying analysis, which we believe still offers a useful benchmark-style resource to the community.

---

> ### Author Response · Authors · 2025-11-27
> **Response for W3**
>
> ## W3: The review is largely a catalog of models with limited critical analysis, offering little insight into why architectures succeed or fail or what principles unify MLLMs for science.
>
> We thank the reviewer for this valuable observation. While Sections 2–5 focus on organizing and comparing existing MLLMs, our analysis is not limited to cataloging models. In **Appendix C: *Emerging Hot Topics and Future Directions***, we provide **deep, cross-domain critical analysis** that directly addresses why certain architectures succeed or fail and what principles unify MLLMs for science.
>
> Specifically:
>
> - **We analyze emerging architectural paradigms**—especially diffusion LLMs (dLLMs) and diffusion MLLMs (dMLLMs)—explaining *how and why* their iterative mask–denoise mechanisms outperform autoregressive models in reasoning, controllability, structure, and global coherence.
>   This includes discussion of decoding dynamics, training variance, scaling strategies, efficiency trade-offs, safety risks, and alignment challenges.
>
> - **We identify fundamental design principles** shared across scientific domains, such as the role of physical constraints, global consistency, multimodal alignment objectives, and structure-aware decoding. We highlight how these principles differ from conventional NLP-oriented LLMs.
>
> - **We provide domain-specific failure modes and challenges** in molecular science, protein modeling, materials discovery, and genomics—covering issues such as physical-law violations, dynamics modeling gaps, long-range dependency capture, multimodal data inconsistencies, and safety limitations.
>
> - **We outline forward-looking research directions**, including quantum-aware molecular representations, protein dynamics integration, multiscale materials modeling, ontology-grounded genomic reasoning, and diffusion-native safety mechanisms—all grounded in unmet scientific needs rather than model enumeration.
>
> Together, these discussions move far beyond listing models:
> they synthesize *architectural insights*, *failure analyses*, and *cross-domain unifying principles* that guide the next generation of MLLMs for scientific discovery.
>
> We appreciate the reviewer’s feedback and have revised the section heading and transitions to make this analytical contribution more prominent.

---

### Official Review · Reviewer_nZJe · 2025-10-31

**Soundness:** 2
**Presentation:** 3
**Contribution:** 1
**Rating:** 0
**Confidence:** 4

**Summary:**

The work does not present any new methodology or dataset but looks like a survey paper mainly focusing on existing model's metadata such as parameter count, release date, modalities trained on etc. As such I personally don't find it suitable for ICLR.

**Strengths:**

N/A

**Weaknesses:**

N/A

**Questions:**

N/A

---

> ### Author Response · Authors · 2025-11-23
> **Sincerely thank the reviewer for the feedback~**
>
> We sincerely thank the reviewer for the feedback. Below we provide brief and respectful responses to the points raised:
>
> ## 1. Our paper is not only a metadata survey — it is a survey plus a reproducible benchmark.
>
> - **Molecular property prediction (MoleculeNet: BACE / BBBP / HIV).**
>    As shown in **Table D1 (Appendix D)**, we evaluate **19 models**, including 8 specialist models (ChemBERTa v2, DMP(TF+GNN), KV-PLM, GraphCL, GraphMVP-C, MoMu, MolFM, Uni-Mol) and 11 LLM/MLLM/generalist models (Galactica-6.7B, Vicuna-v1.5-13b-16k, Vicuna-v1.3-7B, LLaMA-2-7B-chat, MolCA(1D), MolCA(1D+2D), Instruct-G, Instruct-GS, MoleculeSTM(Graph), MoleculeSTM(Smiles), Token-Mol).
>    All results are reported as AUROC across the three MoleculeNet datasets. We also include clear documentation of limitations and reproducibility in the appendix.
> - **Protein property prediction (TAPE suite, 6 tasks).**
>    As summarized in **Table D2 (Appendix D)**, we benchmark **8 representative models** — LSTM, TAPE Transformer, ResNet, MSA Transformer, ProtBERT, OntoProtein, ProteinDT-ProteinCLAP-InfoNCE, and ProteinDT-ProteinCLAP-EBM-NCE — covering six standard protein property prediction tasks (SS-Q3/Q8, Homology, Contact, Fluorescence, Stability) with their standard metrics (accuracy, Top-L/2 precision, Spearman ρ).
>
> Beyond these experiments, our paper also provides structured and comprehensive cross-domain summaries (**Appendix E**), spanning molecules, proteins, genomics, and materials. Taken together, these components demonstrate that the submission is not merely “a list of model metadata,” but a combined **survey + benchmark** effort with meaningful empirical contributions and fully reproducible protocols.
>
> ## 2. ICLR Main Track Precedent for Survey-Style Contributions
>
> We respectfully note that ICLR has a strong history of accepting **carefully executed surveys** that offer insight or valuable community resources. For example, the ICLR 2025 paper "[(Mis)Fitting Scaling Laws: A Survey of Scaling Law Fitting Techniques in Deep Learning"](https://openreview.net/forum?id=xI71dsS3o4) was explicitly categorized as a survey. This indicates that well-designed survey-style research does fall within ICLR’s intended scope.

---

### Official Review · Reviewer_ABMv · 2025-10-31

**Soundness:** 1
**Presentation:** 3
**Contribution:** 2
**Rating:** 2
**Confidence:** 5

**Summary:**

This paper presents an extensive survey and empirical study of multimodal large language models (MLLMs) in scientific research and systematically consolidates developments across multiple scientific domains—including molecular and drug design, protein science, genomics, and materials science. The authors summarize representative model and methods, domain-specific adaptations, and commonly used datasets, providing ataxonomy of how MLLMs integrate textual, visual, structural, and 2D, 3D geometric modalities for each specific domain. The paper also highlights emerging research frontiers such as diffusion-based MLLMs. Its benchmarking component evaluates selected models (e.g., MoMu, MoleculeSTM, Token-Mol) on two tasks: molecular property and protein function prediction, illustrating comparative strengths and limitations.

**Strengths:**

1. The paper presents a systematic and interdisciplinary survey of multimodal large language models (MLLMs) for scientific discovery, covering four major domains—molecular science, protein science, genomics, and materials science. This breadth of coverage reflects both the maturity and the integrative nature of the review.

2.  The work provides an in-depth examination of diverse modalities involved in MLLMs, including textual sequences, 2D and 3D structural representations, and sequence-to-language transformations. It systematically summarizes representative methods and evolutionary trends within each modality, offering a well-organized reference framework for future research.

3. Beyond summarizing existing architectures, the paper also discusses emerging paradigms, expecially diffusion-based multimodal generation models and analyzes their potential applications across different scientific domains.

**Weaknesses:**

1. My major concern is the limited evaluation task coverage. The title of the benchmark is Scientific Discovery, but the task coverage is very limited. It seems that there is an overclaim about the paper. Although the authors conducted a comprehensive literature review across various subfields, the experimental evaluation is restricted to only two tasks—molecular property prediction and protein function prediction. This narrow focus limits the paper’s contribution to understanding and integrating domain-specific challenges and results. Expanding the benchmarking to include additional representative tasks (e.g., gene function prediction, materials property modeling) would significantly enhance the empirical depth and generalizability of the work.

2. Insufficient analysis of benchmarking results. The paper reports the benchmarking outcomes with only brief descriptions and lacks detailed comparative or diagnostic analysis. It does not examine the reasons behind performance differences among models or analyze design factors such as architecture, modality fusion strategy, or training paradigm. Consequently, the study provides limited insight into task-specific findings. Future work should include more granular comparisons and discuss their respective advantages and limitations across modalities and tasks.

3. Absence of evaluation on leading proprietary models. The current experiments are conducted solely on open-source models, without including comparisons against strong proprietary models such as GPT-5 or Gemini-2.5, which have demonstrated advanced multimodal reasoning capabilities. The lack of such baselines weakens the completeness and external validity of the conclusions, as it remains unclear how these closed-source models would perform or what the potential upper bound of performance might be on such scientific multimodal tasks.

**Questions:**

1. Most existing MLLMs remain focused on text generation, rather than directly producing multimodal outputs such as images or structured visualizations. However, in scientific discovery, non-textual outputs can often be more intuitive and interpretable. Do the authors believe that multimodal output generation (e.g., visual depictions, molecular structures) is necessary for advancing scientific research? What do they consider to be the main challenges in achieving such multimodal outputs?

---

> ### Author Response · Authors · 2025-11-27
> **Thank you very much for your comments~**
>
> Thank you very much for your constructive comments and question. Below, we provide a point-by-point response.
>
> ## W1. Limited evaluation coverage
>
> Thank you very much for these constructive suggestions. We appreciate the reviewer’s point that adding tasks from materials science and genomics, as well as a more systematic evaluation of generative capabilities, would broaden the scope of the benchmark.
>
> In the current work, however, we intentionally focus on molecules and proteins, and primarily on property prediction tasks. Extending the benchmark to (i) materials and genomics, and (ii) generation-oriented tasks in all these domains would require substantial additional data curation, domain-specific evaluation design (e.g., stability/validity/novelty metrics for generation), and extensive analysis. In our view, a truly comprehensive benchmark that covers both prediction and generation for molecules, proteins, materials, and genomics would already constitute one or several dedicated benchmark papers, which is beyond the scope of the present submission and not feasible within the rebuttal and revision period.
>
> ## W2. Insufficient analysis of benchmarking results
>
> We agree that this point is crucial. In response, we have added a dedicated subsection that analyzes the experimental results in greater detail, aiming to explain the performance differences among models by relating them to their respective advantages and limitations across modalities and tasks.
> However, it's important to clarify the design of our survey: for each modality, the content is organized task-wise rather than model-wise. This means that for a given task, the involved models may appraoch the problem from different directions or follow different techincal routes, rather than evolving along a single line of methodological improvements. Additionally, many MLLMs in scintific domains are not developed for a single specific task. Their capabilities(e.g., protein understanding) are often evaluated using foundational tasks such as property prediction or function predicition. Our focus is therefore on the intended purpose and design motivations behind each model, which explains why our initial submission did not include an extensive performance-centric analysis.
>
> ## W3. Absence of evaluation on leading proprietary models
>
> We kindly ask for your understanding on this point. Our work specifically focuses on domain-specialized MLLMs for scientific discovery, rather than general-purpose MLLMs. We will make this distinction clearer in the revised manuscript and apologize for any confusion caused by this earlier lack of clarification.

---

> ### Author Response · Authors · 2025-11-27
> **Response for the question**
>
> ## Q1. Does scientific discovery require models to generate visual or structured multimodal outputs, and what are the main challenges in achieving this?
>
> Thank you for this very insightful question.
>
> In our view, the ability for models to generate visual or structured multimodal outputs is clearly important for scientific discovery. Text and 2D images are actually very simplified forms of data. They only capture a small part of how the real physical world works. Even if a model is extremely large, if it only learns from text and images, it cannot fully learn the underlying rules of our physical world.
>
> Scientific data are much more complex. At the microscopic level, molecules and proteins live in 3D space and follow strong physical constraints such as SE(3) symmetry, periodicity, and other structural rules. At the macroscopic level, videos and dynamic scenes also contain physical laws like motion, force, and gravity. These types of structured data directly reflect the real physical world. This is exactly why AI for science is meaningful: models should learn these physical regularities from rich, real-world data, just as humans do.
>
> The main challenge is therefore: **how do we design AI methods that can correctly represent and generate data that follow these physical laws?**
>  For example, how to build models that naturally encode SE(3) equivariance for molecules, periodicity for materials, or realistic dynamics for physical scenes. This is not something text-only or image-only models can solve. It requires new architectures and learning frameworks that respect the underlying physics.
>
> We believe this direction—learning and generating structured data that obey physical rules—has huge potential and is one of the most important challenges for AI in scientific discovery, even for AI itself.

---

### Official Review · Reviewer_kGuq · 2025-10-31

**Soundness:** 3
**Presentation:** 3
**Contribution:** 2
**Rating:** 6
**Confidence:** 3

**Summary:**

This paper presents a comprehensive survey of multimodal large language models (MLLMs) applied to molecular, protein, materials, and genomic scientific discovery. The authors catalogue and categorize recent approaches, discuss their conceptual differences and applicability, and highlight emerging trends across these domains.

**Strengths:**

The survey covers a broad set of application domains—molecules, proteins, materials, and genomics—and provides a useful overview of widely used and well-established methods.

The authors make an effort to explain conceptual distinctions between approaches, which helps readers understand trade-offs and design choices at a high level.

The manuscript is well written and structured in the manner typical for high-quality survey papers; it is easy to follow and adequate as an entry point into the field.

**Weaknesses:**

The manuscript would be substantially strengthened by adding more practical, quantitative comparisons and clearer systematization of methods.

The comparisons currently presented (Tables D1, D2 in the Supplementary) cover only a small subset of molecular and protein prediction tasks and omit materials and genomics.Generative capabilities of the surveyed models are not addressed. Many surveyed works report results on established benchmarks (e.g., MOSES, TDC, USPTO for 2D molecular tasks; GEOM-DRUGS, CROSSDOCKED for 3D molecular tasks). Compiling these publicly reported results into summary tables in the Supplementary Materials would provide essential, practitioner-oriented value and make the survey far more actionable.


I recommend a dedicated section that systematizes approaches by how they integrate modalities, for example: text-only LLMs [a], frozen domain-specific encoders [b, c], trainable domain-specific encoders [d, e], and fully custom multimodal architectures. Such a taxonomy will make conceptual differences clearer and simplify comparison across papers.


Separately overview the data representations used in the literature (e.g., SMILES/SELFIES/IUPAC for 2D molecules; atom coordinates/torsional-angle representations for 3D structures; sequence and structural encodings for proteins; genomic tokenizations). For each representation, briefly state strengths and limitations. This will lower the barrier for newcomers and improve the survey’s utility as a reference.

a. BindGPT: A Scalable Framework for 3D Molecular Design via Language Modeling and Reinforcement Learning, Zholus et al.

b. 3D-MOLM: Towards 3D Molecule-Text Interpretation in Language Models, Li et al.

c. Structure Language Models for Protein Conformation Generation, Lu et al.

d. nach0: Multimodal Natural and Chemical Languages Foundation Model, Kuznetsov et al.

e. 3DSMILES-GPT: 3D molecular pocket-based generation with token-only large language model, Wang et al.

**Questions:**

All important questions and suggestions are stated in Weaknesses section.

Minor:
Typo on lines 74–75: evluation → evaluation.

---

> ### Author Response · Authors · 2025-11-23
> **Thank you very much for your comments~**
>
> We sincerely thank the reviewer for the constructive feedback. Below we provide brief point-by-point responses to each comment.
>
> ## 1. Need for more practical, quantitative comparisons and clearer systematization of methods
> **Response:**
> Thank you for the insightful suggestion. In the revised version, we will include additional quantitative comparisons and introduce a dedicated section that systematically categorizes existing methods based on their modality-integration strategies.
>
> ## 2. Limited task coverage in Tables D1/D2; missing materials/genomics; missing generative capabilities; suggestion to summarize benchmark results
> **Response:**
> Thank you very much for these constructive suggestions. We appreciate the reviewer’s point that adding tasks from materials science and genomics, as well as a more systematic evaluation of generative capabilities, would broaden the scope of the benchmark.
>
> In the current work, however, we intentionally focus on molecules and proteins, and primarily on property prediction tasks. Extending the benchmark to (i) materials and genomics, and (ii) generation-oriented tasks in all these domains would require substantial additional data curation, domain-specific evaluation design (e.g., stability/validity/novelty metrics for generation), and extensive analysis. In our view, a truly comprehensive benchmark that covers both prediction and generation for molecules, proteins, materials, and genomics would already constitute one or several dedicated benchmark papers, which is beyond the scope of the present submission and not feasible within the rebuttal and revision period.
>
> ## 3. Suggestion to add a section categorizing modality-integration approaches
> **Response:**
> We have added a dedicated section comparing major modality-integration paradigms such as text-only LLMs, frozen encoders, trainable encoders, and custom multimodal architectures.
>
> ## 4. Suggestion to discuss data representations and their strengths/limitations
> **Response:**
> We have included a subsection reviewing key representations (SMILES/SELFIES/IUPAC, 3D coordinates, protein sequence/structure encodings, genomic tokenizations) and outline their advantages and limitations.
>
> ## 5. Additional reference recommendations
> **Response:**
> We have incorporated and discuss the suggested references (BindGPT, 3D-MOLM, Structure LM, nach0, 3DSMILES-GPT) in the revised manuscript.
>
> ## 6. Typographical error
> **Response:**
> The typo (“evluation”) has been corrected.

---

### Author Response · Authors · 2025-12-01
**General Response**

We thank all reviewers for their time. constructive feedback, and encouraging assessment of our work. In response to the valuable comments, we have revised the paper accordingly(changes highlighted in **red**). A summary of the key updates is provided below:

- **Title revision**：To avoid the confusion caused by the original use of "benchmark" and to better reflect the nature of our contribution, We have updated the title to: **A Comprehensive Survey of Multimodal LLMs for Scientific Discovery**
(Reviewer e7Ab)

- **Modality integration**: In our introduction to MLLMs, we added a dedicated discussion on how general-purpose MLLMs fuse different modalities, progressively improving their cross-modal understanding abilities. Moreover, although our overview of MLLMs in scientific discovery is organized by tasks-domains, each section on molecules, proteins, genomes, and materials explicitly discusses the modality integration strategies used within that particular field. Thus, modality integration is covered both at the genral MLLM and within each scientific domain(Reviewer kGuq)

- **Data representations**: Since data representations are fundamental to modeling decisions, we have included a new subsection reviewing key representations (SMILES/SELFIES/IUPAC, 3D coordinates, protein sequence/structure encodings, genomic tokenizations) and outline their advantages and limitations.(Reviewer kGuq)

- **Analysis of benchmarking results**: Based on our existing experiments on moleculer property prediction and protein property prediction, we have added a dedicated subsection providing a more detailed analysis of the experimental results, aiming to explain the performance differences among models by relating them to their respective advantages and limitations across modalities and tasks. (Reviewer ABMv)

We also reiterate that our work provides the first comprehensive survey that systematically covers MLLMs across molecules, proteins, genomes, and materials, along with unified taxonomy, modality integration perspectives, and comparative benchmarking analyses in molecules and proteins.

We would like to express our deepest gratitude for the time and effort you and the review team have invested in this review process and hope our clarifications and new results address all remaining concerns. We understand everyone is busy, but we would appreciate it if the reviewers could kindly provide any responses to our rebuttal when convenient. Thank you.

---

### Meta-Review · Area_Chair_f8GA · 2026-01-06

**Summary:**

The paper presents a comprehensive review of recent advances in MLLMs in key scientific domains. Reviewers have concerns on
(1) the use the term "benchmark" in the paper title without introducing new benchmarks, (2) limited experimental study on small subset of molecular and protein prediction tasks while omitting materials and genomics int he study, (3) insufficient analysis of benchmark results, (4) missing evaluation on leading proprietary models such as GPT 5 and Gemini 2.5. The author responses only partially addressed the concerns. Concerns 2-4 are not addressed adequately. Therefore the paper is recommended to be rejected.

**Reviewer Concerns:**

The authors addressed the concerns on adding paragraphs to clarifying modality integration, revising the title, adding missing references, but failed to addressed concerns 2-4 listed above.

**Reviewer Scores:**

The scores would remain unchanged.

---

### Decision · Program_Chairs · 2026-01-26

Reject